# Learning Multiple Markov Chains
# via Adaptive Allocation

**Mohammad Sadegh Talebi**
SequeL Team, Inria Lille – Nord Europe
sadegh.talebi@inria.fr

**Odalric-Ambrym Maillard**
SequeL Team, Inria Lille – Nord Europe
odalric.maillard@inria.fr

## Abstract

We study the problem of learning the transition matrices of a set of Markov chains from a single stream of observations on each chain. We assume that the Markov chains are ergodic but otherwise unknown. The learner can sample Markov chains sequentially to observe their states. The goal of the learner is to sequentially select various chains to learn transition matrices uniformly well with respect to some loss function. We introduce a notion of loss that naturally extends the squared loss for learning distributions to the case of Markov chains, and further characterize the notion of being *uniformly good* in all problem instances. We present a novel learning algorithm that efficiently balances *exploration* and *exploitation* intrinsic to this problem, without any prior knowledge of the chains. We provide finite-sample PAC-type guarantees on the performance of the algorithm. Further, we show that our algorithm asymptotically attains an optimal loss.

## 1 Introduction

We study a *variant* of the following sequential adaptive allocation problem: A learner is given a set of $K$ arms, where to each arm $k \in [K]$, an unknown real-valued distributions $\nu_k$ with mean $\mu_k$ and variance $\sigma_k^2 > 0$ is associated. At each round $t \in \mathbb{N}$, the learner must select an arm $k_t \in [K]$, and receives a sample drawn from $\nu_k$. Given a total budget of $n$ pulls, the objective is to estimate the expected values $(\mu_k)_{k \in [K]}$ of all distributions uniformly well. The quality of estimation in this problem is classically measured through the *expected* quadratic estimation error $\mathbb{E}[(\mu_k - \hat{\mu}_{k,n})^2]$ for the empirical mean estimate $\hat{\mu}_{k,n}$ built with the $T_{k,n} = \sum_{t=1}^{n} \mathbb{I}\{k = k_t\}$ many samples received from $\nu_k$ at time $n$, and the performance of an allocation strategy is the maximal error, $\max_{k \in [K]} \mathbb{E}[(\mu_k - \hat{\mu}_{k,n})^2]$. Using ideas from the Multi-Armed Bandit (MAB) literature, previous works (e.g., [1, 2]) have provided optimistic sampling strategies with near-optimal performance guarantees for this setup.

This generic adaptive allocation problem is related to several application problems arising in optimal experiment design [3, 4], active learning [5], or Monte-Carlo methods [6]; we refer to [1, 7, 2, 8] and references therein for further motivation. We extend this line of work to the case where each process is a *discrete Markov chain*, hence introducing the problem of *active bandit learning of Markov chains*. More precisely, we no longer assume that $(\nu_k)_k$ are real-valued distributions, but we study the case where each $\nu_k$ is a discrete Markov process over a state space $\mathcal{S} \subset \mathbb{N}$. The law of the observations $(X_{k,i})_{i \in \mathbb{N}}$ on arm (or chain) $k$ is given by $\nu_k(X_{k,1}, \ldots X_{k,n}) = p_k(X_{k,1}) \prod_{i=2}^{n} P_k(X_{k,i-1}, X_{k,i})$, where $p_k$ denotes the initial distribution of states, and $P_k$ is the transition function of the Markov chain. The goal of the learner is to learn the *transition matrices* $(P_k)_{k \in [K]}$ uniformly well on the chains. Note that the chains are *not* controlled (we only decide which chain to advance, not the states it transits to).

Before discussing the challenges of the extension to Markov chains, let us give further comments on the performance measure considered in bandit allocation for real-valued distributions: Using the expected quadratic estimation error on each arm $k$ makes sense since when $T_{k,n}, k \in [K]$ are

deterministic, it coincides with $\sigma_k^2/T_{k,n}$, thus suggesting to pull arm $k$ proportionally to its variance $\sigma_k^2$. However, for a learning strategy, $T_{k,n}$ typically depends on all past observations. The presented analyses in these series of works rely on Wald's second identity as the technical device, heavily relying on the use of a quadratic loss criterion, which prevents one from extending the approach therein to other distances. Another peculiarity arising in working with expectations is the order of "max" and "expectation" operators. While it makes more sense to control the *expected value of the maximum*, the works cited above look at *maximum of the expected value*, which is more in line with a pseudo-loss definition rather than the loss; actually in extensions of these works a pseudo-loss is considered instead of this performance measure. As we show, all of these difficulties can be avoided by resorting to a *high probability* setup. Hence, in this paper, we deviate from using an *expected* loss criterion, and rather use a high-probability control. We formally define our performance criterion in Section 2.3.

## 1.1 Related Work

On the one hand, our setup can be framed into the line of works on active bandit allocation, considered for the estimation of reward distributions in MABs as introduced in [1, 7], and further studied in [2, 9]. This has been extended to stratified sampling for Monte-Carlo methods in [10, 8], or to continuous mean functions in, e.g., [11]. On the other hand, our extension from real-valued distributions to Markov chains can be framed into the rich literature on Markov chain estimation; see, e.g., [12, 13, 14, 15, 16, 17]. This stream of works extends a wide range of results from the i.i.d. case to the Markov case. These include, for instance, the law of large numbers for (functions of) state values [17], the central limit theorem for Markov sequences [13] (see also [17, 18]), and Chernoff-type or Bernstein-type concentration inequalities for Markov sequences [19, 20]. Note that the majority of these results are available for *ergodic* Markov chains.

Another stream of research on Markov chains, which is more relevant to our work, investigates learning and estimation of the *transition matrix* (as opposed to its full law); see, e.g., [16, 15, 21, 22]. Among the recent studies falling into this category, [22] investigates learning of the transition matrix with respect to a loss function induced by $f$-divergences in a *minimax setup*, thus extending [23] to the case of Markov chains. [21] derives a Probably Approximately Correct (PAC) type bound for learning the transition matrix of an ergodic Markov chain with respect to the *total variation* loss. It further provides a matching lower bound. Among the existing literature on learning Markov chains, to the best of our knowledge, [21] is the closest to ours. There are however two aspects distinguishing our work: Firstly, the challenge in our problem resides in dealing with *multiple* Markov chains, which is present neither in [21] nor in the other studies cited above. Secondly, our notion of loss does not coincide with that considered in [21], and hence, the lower bound of [21] does not apply to our case.

Among the results dealing with multiple chains, we may refer to learning in the Markovian bandits setup [24, 25, 26]. Most of these studies address the problem of reward maximization over a finite time horizon. We also mention that in a recent study, [27] introduces the so-called active exploration in Markov decision processes, where the transition kernel is *known*, and the goal is rather to learn the mean reward associated to various states. To the best of our knowledge, none of these works address the problem of learning the *transition matrix*. Last, as we target high-probability performance bounds (as opposed to those holding in expectation), our approach is naturally linked to the PAC analysis. [28] provides one of the first PAC bounds for learning discrete distributions. Since then, the problem of learning discrete distributions has been well studied; see, e.g., [29, 30, 23] and references therein. We refer to [23] for a rather complete characterization of learning distribution in a *minimax* setting under a big class of smooth loss functions. We remark that except for very few studies (e.g., [29]), most of these results are provided for discrete distributions.

## 1.2 Overview and Contributions

Our contributions are the following: (i) For the problem of learning Markov chains, we consider a notion of loss function, which appropriately extends the loss function for learning distributions to the case of Markov chains. Our notion of loss is similar to that considered in [22] (we refer to Section 2.3 for a comparison between our notion and the one in [22]). In contrast to existing works on similar bandit allocation problems, our loss function avoids technical difficulties faced when extending the squared loss function to this setup. We further characterize the notion of a "uniformly good algorithm" under the considered loss function for ergodic chains; (ii) We present an *optimistic* algorithm, called **BA-MC**, for active learning of Markov chains, which is simple to

implement and does not require any prior knowledge of the chains. To the best of our knowledge, this constitutes the first algorithm for active bandit allocation for learning Markov chains; (iii) We provide non-asymptotic PAC-type bounds as well as an asymptotic one on the loss incurred by **BA-MC**, indicating three regimes. In the first regime, which holds for any learning budget $n \geq 4K$, we present (in Theorem 1) a high-probability bound on the loss scaling as $\widetilde{\mathcal{O}}(\frac{KS^2}{n})$, where $\widetilde{\mathcal{O}}(\cdot)$ hides $\log(\log(n))$ factors. Here, $K$ and $S$ respectively denote the number of chains and the number of states in a given chain. This result holds for homogenous Markov chains. We then characterize a cut-off budget $n_{\text{cutoff}}$ (in Theorem 2) so that when $n \geq n_{\text{cutoff}}$, the loss behaves as $\widetilde{\mathcal{O}}(\frac{\Lambda}{n} + \frac{C_0}{n^{3/2}})$, where $\Lambda = \sum_k \sum_{x,y} P_k(x,y)(1 - P_k(x,y))$ denotes the sum of variances of all states and all chains, and where $P_k$ denotes the transition probability of chain $k$. This latter bound constitutes the second regime, in view of the fact that $\frac{\Lambda}{n}$ equals the asymptotically optimal loss (see Section 2.4 for more details). Thus, this bound indicates that the *pseudo-excess loss* incurred by the algorithm vanishes at a rate $C_0 n^{-3/2}$ (we refer to Section 4 for a more precise definition). Furthermore, we carefully characterize the constant $C_0$. In particular, we discuss that $C_0$ does not deteriorate with mixing times of the chains, which, we believe, is a strong feature of our algorithm. We also discuss how various properties of the chains, e.g., discrepancies between stationary distribution of various states of a given chain, may impact the learning performance. Finally, we demonstrate a third regime, the asymptotic one, when the budget $n$ grows large, in which we show (in Theorem 3) that the loss of **BA-MC** matches the asymptotically optimal loss $\frac{\Lambda}{n}$. All proofs are provided in the supplementary material.

Markov chains have been successfully used for modeling a broad range of practical problems, and their success makes the studied problem in this paper relevant in practice. There are practical applications in reinforcement learning (e.g., active exploration in MDPs [27]) and in rested Markov bandits (e.g., channel allocation in wireless communication systems where a given channel's state follows a Markov chain[1]), for which we believe our contributions could serve as a technical tool.

## 2 Preliminaries and Problem Statement

### 2.1 Preliminaries

Before describing our model, we recall some preliminaries on Markov chains; these are standard definitions and results, and can be found in, e.g., [32, 33]. Consider a Markov chain defined on a finite state space $\mathcal{S}$ with cardinality $S$. Let $\mathcal{P}_{\mathcal{S}}$ denote the collection of all row-stochastic matrices over $\mathcal{S}$. The Markov chain is specified by its transition matrix $P \in \mathcal{P}_{\mathcal{S}}$ and its initial distribution $p$: For all $x, y \in \mathcal{S}$, $P(x, y)$ denotes the probability of transition to $y$ if the current state is $x$. In what follows, we may refer to a chain by just referring to its transition matrix.

We recall that a Markov chain $P$ is *ergodic* if $P^m > 0$ (entry-wise) for some $m \in \mathbb{N}$. If $P$ is ergodic, then it has a unique *stationary distribution* $\pi$ satisfying $\pi = \pi P$. Moreover $\underline{\pi} := \min_{x \in \mathcal{S}} \pi(x) > 0$. A chain $P$ is said to be *reversible* if its stationary distribution $\pi$ satisfies *detailed balance equations*: For all $x, y \in \mathcal{S}$, $\pi(x)P(x, y) = \pi(y)P(y, x)$. Otherwise, $P$ is called *non-reversible*. For a Markov chain $P$, the largest eigenvalue is $\lambda_1(P) = 1$ (with multiplicity one). In a *reversible* chain $P$, all eigenvalues belong to $(-1, 1]$. We define the *absolute spectral gap* of a reversible chain $P$ as $\gamma(P) = 1 - \lambda_\star(P)$, where $\lambda_\star(P)$ denotes the second largest (in absolute value) eigenvalue of $P$. If $P$ is reversible, the absolute spectral gap $\gamma(P)$ controls the convergence rate of the state distributions of the chain towards the stationary distribution $\pi$. If $P$ is *non-reversible*, the convergence rate is determined by the *pseudo-spectral gap* as introduced in [20] as follows. Define $P^\star$ as: $P^\star(x, y) := \pi(y)P(y, x)/\pi(x)$ for all $x, y \in \mathcal{S}$. Then, the pseudo-spectral gap $\gamma_{\mathsf{ps}}(P)$ is defined as: $\gamma_{\mathsf{ps}}(P) := \max_{\ell \geq 1} \frac{\gamma((P^\star)^\ell P^\ell)}{\ell}$.

### 2.2 Model and Problem Statement

We are now ready to describe our model. We consider a learner interacting with a finite set of Markov chains indexed by $k \in [K] := \{1, 2, \ldots, K\}$. For ease of presentation, we assume that all Markov chains are defined on the same state space[2] $\mathcal{S}$ with cardinality $S$. The Markov chain $k$, or for short

chain $k$, is specified by its transition matrix $P_k \in \mathcal{P}_\mathcal{S}$. In this work, we assume that all Markov chains are ergodic, which implies that any chain $k$ admits a unique stationary distribution, which we denote by $\pi_k$. Moreover, the minimal element of $\pi_k$ is bounded away from zero: $\underline{\pi}_k := \min_{x \in \mathcal{S}} \pi_k(x) > 0$. The initial distributions of the chains are assumed to be arbitrary. Further, we let $\gamma_k := \gamma(P_k)$ denote the absolute spectral gap of chain $k$ if $k$ is reversible; otherwise, we define the pseudo-spectral gap of $k$ by $\gamma_{\mathsf{ps},k} := \gamma_{\mathsf{ps}}(P_k)$.

A related quantity in our results is the *Gini index* of the various states. For a chain $k$, the *Gini index* for state $x \in \mathcal{S}$ is defined as

$$G_k(x) := \sum_{y \in \mathcal{S}} P_k(x,y)(1 - P_k(x,y)).$$

Note that $G_k(x) \leq 1 - \frac{1}{S}$. This upper bound is verified by the fact that the maximal value of $G_k(x)$ is achieved when $P_k(x,y) = \frac{1}{S}$ for all $y \in \mathcal{S}$ (in view of the concavity of $z \mapsto \sum_{x \in \mathcal{S}} z(x)(1 - z(x))$). In this work, we assume that for all $k$, $\sum_{x \in \mathcal{S}} G_k(x) > 0$.[3] Another related quantity in our results is the sum (over states) of inverse stationary distributions: For a chain $k$, we define $H_k := \sum_{x \in \mathcal{S}} \pi_k(x)^{-1}$. Note that $S^2 \leq H_k \leq S\underline{\pi}_k^{-1}$. The quantity $H_k$ reflects the discrepancy between individual elements of $\pi_k$.

**The online learning problem.** The learner wishes to design a sequential allocation strategy to adaptively sample various Markov chains so that all transition matrices are learnt uniformly well. The game proceeds as follows: Initially all chains are assumed to be non-stationary with arbitrary initial distributions chosen by the environment. At each step $t \geq 1$, the learner samples a chain $k_t$, based on the past decisions and the observed states, and observes the state $X_{k_t,t}$. The state of $k_t$ evolves according to $P_{k_t}$. The state of chains $k \neq k_t$ does not change: $X_{k,t} = X_{k,t-1}$ for all $k \neq k_t$.

We introduce the following notations: Let $T_{k,t}$ denote the number of times chain $k$ is selected by the learner up to time $t$: $T_{k,t} := \sum_{t'=1}^{t} \mathbb{I}\{k_{t'} = k\}$, where $\mathbb{I}\{\cdot\}$ denotes the indicator function. Likewise, we let $T_{k,x,t}$ represent the number of observations of chain $k$, up to time $t$, when the chain was in state $x$: $T_{k,x,t} := \sum_{t'=1}^{t} \mathbb{I}\{k_{t'} = k, X_{k,t'} = x\}$. Further, we note that the learner only controls $T_{k,t}$ (or equivalently, $\sum_x T_{k,x,t}$), but not the number of visits to individual states. At each step $t$, the learner maintains empirical estimates of the stationary distributions, and estimates transition probabilities of various chains based on the observations gathered up to $t$. We define the empirical stationary distribution of chain $k$ at time $t$ as $\hat{\pi}_{k,t}(x) := T_{k,x,t}/T_{k,t}$ for all $x \in \mathcal{S}$. For chain $k$, we maintain the following *smoothed estimation* of transition probabilities:

$$\widehat{P}_{k,t}(x,y) := \frac{\alpha + \sum_{t'=2}^{t} \mathbb{I}\{X_{k,t'-1} = x, X_{k,t'} = y\}}{\alpha S + T_{k,x,t}}, \quad \forall x, y \in \mathcal{S}, \tag{1}$$

where $\alpha$ is a positive constant. In the literature, the case of $\alpha = \frac{1}{S}$ is usually referred to as the *Laplace-smoothed* estimator. The learner is given a budget of $n$ samples, and her goal is to obtain an accurate estimation of transition matrices of the Markov chains. The accuracy of the estimation is determined by some notion of loss, which will be discussed later. The learner adaptively selects various chains so that the minimal loss is achieved.

## 2.3 Performance Measures

We are now ready to provide a precise definition of our notion of loss, which would serve as the performance measure of a given algorithm. Given $n \in \mathbb{N}$, we define the loss of an adaptive algorithm $\mathcal{A}$ as:

$$L_n(\mathcal{A}) := \max_{k \in [K]} L_{k,n}, \quad \text{with} \quad L_{k,n} := \sum_{x \in \mathcal{S}} \hat{\pi}_{k,n}(x) \| P_k(x,\cdot) - \widehat{P}_{k,n}(x,\cdot) \|_2^2.$$

The use of the $L_2$-norm in the definition of loss is quite natural in the context of learning and estimation of distributions, as it is directly inspired by the quadratic estimation error used in active

bandit allocation (e.g., [2]). Given a budget $n$, the loss $L_n(\mathcal{A})$ of an adaptive algorithm $\mathcal{A}$ is a random variable, due to the evolution of the various chains as well as the possible randomization in the algorithm. Here, we aim at controlling this random quantity in a *high probability* setup as follows: Let $\delta \in (0, 1)$. For a given algorithm $\mathcal{A}$, we wish to find $\varepsilon := \varepsilon(n, \delta)$ such that

$$\mathbb{P}\left(L_n(\mathcal{A}) \geq \varepsilon\right) \leq \delta. \tag{2}$$

**Remark 1** *We remark that the empirical stationary distribution $\hat{\pi}_{k,t}$ may differ from the stationary distribution associated to the smoothed estimator $\widehat{P}_{k,t}$ of the transition matrix. Our algorithm and results, however, do not rely on possible relations between $\hat{\pi}_{k,t}$ and $\widehat{P}_{k,t}$, though one could have used smoothed estimators for $\pi_k$. The motivation behind using empirical estimate $\hat{\pi}_{k,t}$ of $\pi_k$ in $L_n$ is that it naturally corresponds to the occupancy of various states according to a given sample path.*

**Comparison with other losses.** We now turn our attention to the comparison between our loss function and some other possible notions. First, we compare ours to the loss function $L'_n(\mathcal{A}) = \max_k \sum_{x \in \mathcal{S}} \|P_k(x, \cdot) - \widehat{P}_{k,n}(x, \cdot)\|_2^2$. Such a notion of loss might look more natural or simpler, since the weights $\hat{\pi}_{k,n}(x)$ are replaced simply with 1 (equivalently, uniform weights). However, this means a strategy may incur a high loss for a part of the state space that is rarely visited, even though we have absolutely no control on the chain. For instance, in the extreme case when some states $x$ are reachable with a very small probability, $T_{k,x,n}$ may be arbitrarily small thus resulting in a large loss $L'_n$ for all algorithms, while it makes little sense to penalize an allocation strategy for these "virtual" states. Weighting the loss according to the empirical frequency $\hat{\pi}_{k,n}$ of visits avoids such a phenomenon, and is thus more meaningful.

In view of the above discussion, it is also tempting to replace the empirical state distribution $\hat{\pi}_{k,n}$ with its expectation $\pi_k$, namely to define a *pseudo-loss* function of the form $L''_n(\mathcal{A}) = \max_k \sum_x \pi_k(x)\|P_k(x, \cdot) - \widehat{P}_{k,n}(x, \cdot)\|_2^2$ (as studied in, e.g., [22] in a different setup). We recall that our aim is to derive performance guarantees on the algorithm's loss that hold with high probability (for $1 - \delta$ portions of the sample paths of the algorithm for a given $\delta$). To this end, $L_n$ (which uses $\hat{\pi}_{k,n}$) is more natural and meaningful than $L''_n$ as $L_n$ penalizes the algorithm's performance by the relative visit counts of various states in a given sample path (through $\hat{\pi}_{k,n}$), and not by the expected value of these. This matters a lot in the small-budget regime, where $\hat{\pi}_{k,n}$ could differ significantly from $\pi_k$ — Otherwise when $n$ is large enough, $\hat{\pi}_{k,n}$ becomes well-concentrated around $\pi_k$ with high probability. To clarify further, let us consider the small-budget regime, and some state $x$ where $\pi_k(x)$ is not small. In the case of $L_n$, using $\hat{\pi}_{k,n}$ we penalize the performance by the mismatch between $\widehat{P}_{k,n}(x, \cdot)$ and $P_k(x, \cdot)$, weighted proportionally to the number of rounds the algorithm has actually visited $x$. In contrast, in the case of $L''_n$, weighting the mismatch proportionally to $\pi_k(x)$ does not seem reasonable since in a given sample path, the algorithm might not have visited $x$ enough even though $\pi_k(x)$ is not small. We remark that our results in subsequent sections easily apply to the pseudo-loss $L''_n$, at the expense of an additive second-order term, which might depend on the mixing times.

Finally, we position the high-probability guarantee on $L_n$, in the sense of Eq. (2), against those holding in expectation. Prior studies on bandit allocation, such as [7, 2], whose objectives involve a max operator, consider expected squared distance. The presented analyses in these series of works rely on Wald's second identity as the technical device. This prevents one to extend the approach therein to other distances. Another peculiarity arising in working with expectations is the order of "max" and "expectation" operators. While it makes more sense to control the *expected value of the maximum*, the works cited above look at *maximum of the expected value*, which is more in line with a pseudo-loss definition rather than the loss. All of these difficulties can be avoided by resorting to a high probability setup (in the sense of Eq. (2).

**Further intuition and example.** We now provide an illustrative example to further clarify some of the above comments. Let us consider the following two-state Markov chain: $P = \begin{bmatrix} 1/2 & 1/2 \\ \varepsilon & 1 - \varepsilon \end{bmatrix}$,

where $\varepsilon \in (0, 1)$. The stationary distribution of this Markov chain is $\pi = [\frac{\varepsilon}{2+\varepsilon}, \frac{2}{2+\varepsilon}]$. Let $s_1$ (resp. $s_2$) denote the state corresponding to the first (resp. second) row of the transition matrix. In view of $\pi$, when $\varepsilon \ll 1$, the chain tends to stay in $s_2$ (the lazy state) most of the time: Out of $n$ observations, one gets on average only $n\pi(s_1) = n\varepsilon/(2 + \varepsilon)$ observations of state $s_1$, which means, for $\varepsilon \ll 1/n$, essentially no observation of state $s_1$. Hence, no algorithm can estimate the transitions from $s_1$ in

such a setup, and all strategies would suffer a huge loss according to $L'_n$, no matter how samples are allocated to this chain. Thus, $L'_n$ has limited interest in order to distinguish between good and base sampling strategies. On the other hand, using $L_n$ enables to better distinguish between allocation strategies, since the weight given to $s_1$ would be essentially 0 in this case, thus focusing on the good estimation of $s_2$ (and other chains) only.

## 2.4 Static Allocation

In this subsection, we investigate the optimal loss asymptotically achievable by an oracle policy that is aware of some properties of the chains. To this aim, let us consider a non-adaptive strategy where sampling of various chains is deterministic. Therefore, $T_{k,n}, k = 1, \ldots, K$ are not random. The following lemma is a consequence of the central limit theorem:

**Lemma 1** *We have for any chain $k$: $T_{k,n} L_{k,n} \to_{T_{k,n} \to \infty} \sum_x G_k(x)$.*

The proof of this lemma consists in two steps: First, we provide lower and upper bounds on $L_{k,n}$ in terms of the loss $\widetilde{L}_{k,n}$ incurred by the learner had she used an empirical estimator (corresponding to $\alpha = 0$ in (1)). Second, we show that by the central limit theorem, $T_{k,n} \widetilde{L}_{k,n} \to_{T_{k,n} \to \infty} \sum_x G_k(x)$.

Now, consider an oracle policy $\mathcal{A}_{\text{oracle}}$, who is aware of $\sum_{x \in \mathcal{S}} G_k(x)$ for various chains. In view of the above discussion, and taking into account the constraint $\sum_{k \in [K]} T_{k,n} = n$, it would be asymptotically optimal to allocate $T_{k,n} = \eta_k n$ samples to chain $k$, where

$$\eta_k := \frac{1}{\Lambda} \sum_{x \in \mathcal{S}} G_k(x), \quad \text{with} \quad \Lambda := \sum_{k \in [K]} \sum_{x \in \mathcal{S}} G_k(x).$$

The corresponding loss would satisfy: $n L_n(\mathcal{A}_{\text{oracle}}) \to_{n \to \infty} \Lambda$. We shall refer to the quantity $\frac{\Lambda}{n}$ as *the asymptotically optimal loss*, which is a *problem-dependent* quantity. The coefficients $\eta_k, k \in [K]$ characterize the discrepancy between the transition matrices of the various chains, and indicate that an algorithm needs to account for such discrepancy in order to achieve the asymptotically optimal loss. Having characterized the notion of asymptotically optimal loss, we are now ready to define the notion of *uniformly good algorithm*:

**Definition 1 (Uniformly Good Algorithm)** *An algorithm $\mathcal{A}$ is said to be* uniformly good *if, for any problem instance, it achieves the asymptotically optimal loss when $n$ grows large; that is, $\lim_{n \to \infty} n L_n(\mathcal{A}) = \Lambda$ for all problem instances.*

## 3 The BA-MC Algorithm

In this section, we introduce an algorithm designed for adaptive bandit allocation of a set of Markov chains. It is designed based on the *optimistic principle*, as in MAB problems (e.g., [34, 35]), and relies on an index function. More precisely, at each time $t$, the algorithm maintains an index function $b_{k,t+1}$ for each chain $k$, which provides an upper confidence bound (UCB) on the loss incurred by $k$ at $t$; more precisely, with high probability, $b_{k,t+1} \geq L_{k,t} := \sum_{x \in \mathcal{S}} \hat{\pi}_{k,t}(x) \| P_k(x, \cdot) - \widehat{P}_{k,t}(x, \cdot) \|_2^2$, where $\widehat{P}_{k,t}$ denotes the smoothed estimate of $P_k$ with some $\alpha > 0$ (see Eq. (1)). Now, by sampling a chain $k_t \in \text{argmax}_{k \in [K]} b_{k,t+1}$ at time $t$, we can balance exploration and exploitation by selecting more the chains with higher estimated losses or those with higher uncertainty in these estimates.

In order to specify the index function $b_{k,\cdot}$, let us choose $\alpha = \frac{1}{3S}$ (we motivate this choice of $\alpha$ later on), and for each state $x \in \mathcal{S}$, define the estimate of Gini coefficient at time $t$ as $\widehat{G}_{k,t}(x) := \sum_{y \in \mathcal{S}} \widehat{P}_{k,t}(x, y) \big( 1 - \widehat{P}_{k,t}(x, y) \big)$. The index $b_{k,t+1}$ is then defined as

$$b_{k,t+1} = \frac{2\beta}{T_{k,t}} \sum_{x \in \mathcal{S}} \mathbb{I}\{T_{k,x,t} > 0\} \widehat{G}_{k,t}(x) + \frac{28\beta^2 S}{T_{k,t}} \sum_{x \in \mathcal{S}} \frac{\mathbb{I}\{T_{k,x,t} > 0\}}{T_{k,x,t} + \alpha S}$$

$$+ \frac{6.6\beta^{3/2}}{T_{k,t}} \sum_{x \in \mathcal{S}} \frac{T_{k,x,t}^{3/2}}{(T_{k,x,t} + \alpha S)^2} \sum_{y \in \mathcal{S}} \sqrt{\widehat{P}_{k,t}(x, y) \big( 1 - \widehat{P}_{k,t}(x, y) \big)},$$

where $\beta := \beta(n, \delta) := c \log \left( \left\lceil \frac{\log(n)}{\log(c)} \right\rceil \frac{6KS^2}{\delta} \right)$, with $c > 1$ being an arbitrary choice. In this paper, we choose $c = 1.1$.

We remark that the design of the index $b_{k,\cdot}$ above comes from the application of empirical Bernstein concentration for $\alpha$-smoothed estimators (see Lemma 4 in the supplementary) to the loss function $L_{k,t}$. In other words, Lemma 4 guarantees that with high probability, $b_{k,t+1} \geq L_{k,t}$. Our concentration inequality (Lemma 4) is new, to our knowledge, and could be of independent interest.

Having defined the index function $b_{k,\cdot}$, we are now ready to describe our algorithm, which we call **BA-MC** (Bandit Allocation for Markov Chains). **BA-MC** receives as input a confidence parameter $\delta$, a budget $n$, as well as the state space $\mathcal{S}$. It initially samples each chain twice (hence, this phase lasts for $2K$ rounds). Then, **BA-MC** simply consists in sampling the chain with the largest index $b_{k,t+1}$ at each round $t$. Finally, it returns, after $n$ pulls, an estimate $\widehat{P}_{k,n}$ for each chain $k$. We provide the pseudo-code of **BA-MC** in Algorithm 1. Note that **BA-MC** does not require any prior knowledge of the chains (neither the initial distribution nor the mixing time).

---

**Algorithm 1 BA-MC** – Bandit Allocation for Markov Chains

> **Input:** Confidence parameter $\delta$, budget $n$, state space $\mathcal{S}$;
> **Initialize:** Sample each chain twice;
> **for** $t = 2K + 1, \ldots, n$ **do**
>     Sample chain $k_t \in \operatorname{argmax}_k b_{k,t+1}$;
>     Observe $X_{k,t}$, and update $T_{k,x,t}$ and $T_{k,t}$;
> **end for**

---

In order to provide more insights into the design of **BA-MC**, let us remark that (as shown in Lemma 8 in the supplementary) $b_{k,t+1}$ provides a high-probability UCB on the quantity $\frac{1}{T_{k,t}} \sum_x G_k(x)$ as well. Now by sampling the chain $k_t \in \operatorname{argmax}_{k \in [K]} b_{k,t+1}$ at time $t$, in view of discussions in Section 2.4, **BA-MC** would try to mimic an oracle algorithm being aware of $\sum_x G_k(x)$ for various chains.

We remark that our concentration inequality in Lemma 4 (of the supplementary) parallels the one presented in Lemma 8.3 in [36]. In contrast, our concentration lemma makes appear the terms $T_{k,x,t} + \alpha S$ in the denominator, whereas Lemma 8.3 in [36] makes appear terms $T_{k,x,t}$ in the denominator. This feature plays an important role to deal with situations where some states are not sampled up to time $t$, that is for when $T_{k,x,t} = 0$ for some $x$.

## 4 Performance Bounds

We are now ready to study the performance bounds on the loss $L_n(\textbf{BA-MC})$ in both asymptotic and non-asymptotic regimes. We begin with a generic non-asymptotic bound as follows:

**Theorem 1 (BA-MC, Generic Performance)** *Let $\delta \in (0, 1)$. Then, for any budget $n \geq 4K$, with probability at least $1 - \delta$, the loss under $\mathcal{A} = \textbf{BA-MC}$ satisfies*

$$L_n(\mathcal{A}) \leq \frac{287KS^2\beta^2}{n} + \widetilde{\mathcal{O}}\left(\frac{K^2 S^2}{n^2}\right).$$

The proof of this theorem, provided in Section C in the supplementary, reveals the motivation to choose $\alpha = \frac{1}{3S}$: It verifies that to minimize the dependency of the loss on $S$, on must choose $\alpha \propto S^{-1}$. In particular, the proof does not rely on the ergodicity assumption:

**Remark 2** *Theorem 1 is valid even if the Markov chains $P_k, k \in [K]$ are reducible or periodic.*

In the following theorem, we state another non-asymptotic bound on the performance of **BA-MC**, which refines Theorem 1 for when $n \geq n_{\text{cutoff}}$, where

$$n_{\text{cutoff}} := n_{\text{cutoff}}(\delta) := K \max_k \left( \frac{300}{\gamma'_k \underline{\pi}_k} \log \left( \frac{2KS}{\delta} \sqrt{\underline{\pi}_k^{-1}} \right) \right)^2,$$

where $\gamma'_k = \gamma_k$ if $k$ is reversible, and $\gamma'_k = \gamma_{\text{ps},k}$ otherwise. To present the theorem, we recall the notation $\Lambda := \sum_k \sum_x G_k(x)$, and that for any chain $k$, $\eta_k = \frac{1}{\Lambda} \sum_{x \in \mathcal{S}} G_k(x)$, $H_k := \sum_{x \in \mathcal{S}} \pi_k(x)^{-1}$, and $\underline{\pi}_k := \min_{x \in \mathcal{S}} \pi_k(x) > 0$.

**Theorem 2** *Let $\delta \in (0, 1)$, and assume that $n \geq n_{cutoff}$. Then, with probability at least $1 - 2\delta$,*

$$L_n(\mathcal{A}) \leq \frac{2\beta\Lambda}{n} + \frac{C_0\beta^{3/2}}{n^{3/2}} + \widetilde{\mathcal{O}}(n^{-2}),$$

*where* $C_0 := 150K\sqrt{S\Lambda \max_k H_k} + 3\sqrt{S\Lambda}\max_k \frac{H_k}{\eta_k}$.

Recalling the asymptotic loss of the oracle algorithm discussed in Section 2.4 being equal to $\Lambda/n$, in view of the Bernstein concentration, the oracle would incur a loss at most $\frac{2\beta\Lambda}{n}$ for when the budget $n$ is finite. In this regard, we may look at the quantity $L_n(\mathcal{A}) - \frac{2\beta\Lambda}{n}$ as the *pseudo-excess loss* of $\mathcal{A}$ (we refrain from calling this quantity the *excess loss* as $\frac{2\beta\Lambda}{n}$ is not equal to the high-probability loss of the oracle). Theorem 2 implies that when $n$ is greater than the cut-off budget $n_{cutoff}$, the pseudo-excess loss under **BA-MC** vanishes at a rate $\widetilde{\mathcal{O}}(n^{-3/2})$. In particular, Theorem 2 characterizes the constant $C_0$ controlling the main term of the pseudo-excess loss: $C_0 = \mathcal{O}(K\sqrt{S\Lambda \max_k H_k} + \sqrt{S\Lambda}\max_k \frac{H_k}{\eta_k})$. This further indicates that the pseudo-excess loss is controlled by the quantity $\frac{H_k}{\eta_k}$, which captures (i) the discrepancy among the $\sum_x G_k(x)$ values of various chains $k$, and (ii) the discrepancy between various stationary probabilities $\pi_k(x), x \in \mathcal{S}$. We emphasize that the dependency of the learning performance (through $C_0$) on $H_k$ is in alignment with the result obtained in [21] for the estimation of a single ergodic Markov chain.

The proof of Theorem 2, provided in Section D in the supplementary, shows that to determine the cut-off budget $n_{cutoff}$, one needs to determine the value of $n$ such that with high probability, for any chain $k$ and state $x$, the term $T_{k,n}(T_{k,x,n} + \alpha S)^{-1}$ approaches $\pi_k(x)^{-1}$, which is further controlled by $\gamma_{\mathsf{ps},k}$ (or $\gamma_k$ if $k$ is reversible) as well as the minimal stationary distribution $\underline{\pi}_k$. This in turn allows us to show that, under **BA-MC**, the number $T_{k,n}$ of samples for any chain $k$ comes close to the quantity $\eta_k n$. Finally, we remark that the proof of Theorem 2 also reveals that the result in the theorem is indeed valid for any constant $\alpha > 0$.

In the following theorem, we characterize the asymptotic performance of **BA-MC**:

**Theorem 3 (BA-MC, Asymptotic Regime)** *Under $\mathcal{A} =$**BA-MC**, $\limsup_{n\to\infty} nL_n(\mathcal{A}) = \Lambda$.*

The above theorem asserts that, asymptotically, the loss under **BA-MC** matches the asymptotically optimal loss $\Lambda/n$ characterized in Section 2.4. We may thus conclude that **BA-MC** is uniformly good (in the sense of Definition 1). The proof of Theorem 3 (provided in Section E of the supplementary) proceeds as follows: It divides the estimation problem into two consecutive sub-problems, the one with the budget $n_0 = \sqrt{n}$ and the other with the rest $n - \sqrt{n}$ of pulls. We then show when $n_0 = \sqrt{n} \geq n_{cutoff}$, the number of samples on each chain $k$ at the end of the first sub-problem is lower bounded by $\Omega(n^{1/4})$, and as a consequence, the index $b_k$ would be accurate enough: $b_{k,n_0} \in \frac{1}{T_{k,n_0}}(\sum_x G_k(x), \sum_x G_k(x) + \widetilde{\mathcal{O}}(n^{-1/8}))$ with high probability. This allows us to relate the allocation under **BA-MC** in the course of the second sub-problem to that of the oracle, and further to show that the difference vanishes as $n \to \infty$.

Below, we provide some further comments about the presented bounds in Theorems 1–3:

**Various regimes.** Theorem 1 provides a non-asymptotic bound on the loss valid for any $n$, while Theorem 3 establishes the optimality of **BA-MC** in the asymptotic regime. In view of the inequality $\Lambda \leq K(S-1)$, the bound in Theorem 1 is at least off by a factor of $S$ from the asymptotic loss $\Lambda/n$. Theorem 2 bridges between the two results thereby establishing a third regime, in which the algorithm enjoys the asymptotically optimal loss up to an additive pseudo-excess loss scaling as $\widetilde{\mathcal{O}}(n^{-3/2})$.

**The effect of mixing.** It is worth emphasizing that the mixing times of the chains *do not* appear explicitly in the bounds, and only control (through the pseudo-spectral gap $\gamma_{\mathsf{ps},k}$) the cut-off budget $n_{cutoff}$ that ensures when the pseudo-excess loss vanishes at a rate $n^{-3/2}$. This is indeed a strong aspect of our results due to our meaningful definition of loss, which could be attributed to the fact that our loss function employs empirical estimates $\hat{\pi}_{k,n}$ in lieu of $\pi_k$. Specifically speaking, as argued in [36], given the number of samples of various states (akin to using $\hat{\pi}_{k,t}(x)$ in the loss definition), the convergence of frequency estimates towards the true values is independent of the mixing time of the chain. We note that despite the dependence of $n_{cutoff}$ on the mixing times, **BA-MC** does not need to

estimate them as when $n \leq n_{\text{cutoff}}$, it still enjoys the loss guarantee of Theorem 1. We also mention that to define an index function for the loss function $\max_k \sum_x \pi_k(x) \|P_k(x, \cdot) - \widehat{P}_{k,n}(x, \cdot)\|_2^2$, one may have to derive confidence bounds on the mixing time and/or stationary distribution $\pi_k$ as well.

**More on the pseudo-excess loss.** We stress that the notion of the pseudo-excess loss bears some similarity with the definition of regret for active bandit learning of distributions as introduced in [7, 2] (see Section 1). In the latter case, the regret typically decays as $n^{-3/2}$ similarly to the pseudo-excess loss in our case. An interesting question is whether the decay rate of the pseudo-excess loss, as a function of $n$, can be improved. And more importantly, if a (problem-dependent) lower bound on the pseudo-excess loss can be established. These questions are open even for the simpler case of active learning of distributions in the i.i.d. setup; see, e.g., [37, 8, 2]. We plan to address these as a future work.

## 5 Conclusion

In this paper, we addressed the problem of active bandit allocation in the case of discrete and ergodic Markov chains. We considered a notion of loss function appropriately extending the loss function for learning distributions to the case of Markov chains. We further characterized the notion of a "uniformly good algorithm" under the considered loss function. We presented an algorithm for learning Markov chains, which we called **BA-MC**. Our algorithm is simple to implement and does not require any prior knowledge of the Markov chains. We provided non-asymptotic PAC-type bounds on the loss incurred by **BA-MC**, and showed that asymptotically, it incurs an optimal loss. We further discussed that the (pseudo-excess) loss incurred by **BA-MC** in our bounds does not deteriorate with mixing times of the chains. As a future work, we plan to derive a problem-dependent lower bound on the pseudo-excess loss. Another interesting, and yet very challenging, future direction is to devise adaptive learning algorithms for restless Markov chains, where the state of various chains evolve at each round independently of the learner's decision.

## Acknowledgements

This work has been supported by CPER Nord-Pas-de-Calais/FEDER DATA Advanced data science and technologies 2015-2020, the French Ministry of Higher Education and Research, Inria, and the French Agence Nationale de la Recherche (ANR), under grant ANR-16-CE40-0002 (project BADASS).

## Footnotes

[1]For example, in the Gilbert-Elliott channels [31].

[2]Our algorithm and results are straightforwardly extended to the case where the Markov chains are defined on different state spaces.

[3]We remark that there exist chains with $\sum_x G_k(x) = 0$. In view of the definition of the Gini index, such chains are necessarily deterministic (or degenerate), namely their transition matrices belong to $\{0,1\}^{S \times S}$. One example is a deterministic cycle with $S$ nodes. We note that such chains may fail to satisfy irreducibility or aperiodicity.

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
