[Supplementary Material]

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

**Notational Convention.** Throughout the appendix, we use the following notational convention: For matrices $A, B \in \mathbb{R}^{m \times m}$ with $m \in \mathbb{N}$, we let $[AB](i,j) := A(i,j)B(i,j)$ for all $i, j \in [m]$.

## A   Concentration Inequalities

**Lemma 2 ([38, Lemma 2.4])** *Let $Z = (Z_t)_{t \in \mathbb{N}}$ be a sequence of random variables generated by a predictable process, and $\mathcal{F} = (\mathcal{F}_t)_t$ be its natural filtration. Let $\varphi : \mathbb{R} \to \mathbb{R}_+$ be a convex upper-envelope of the cumulant generating function of the conditional distributions with $\varphi(0) = 0$, and let $\varphi_\star$ denote its Legendre-Fenchel transform, that is:*

$$\forall \lambda \in \mathcal{D}, \forall t, \qquad \log \mathbb{E}\left[\exp\left(\lambda Z_t\right) | \mathcal{F}_{t-1}\right] \leq \varphi(\lambda) \,,$$

$$\forall x \in \mathbb{R}, \qquad \varphi_\star(x) = \sup_{\lambda \in \mathbb{R}}(\lambda x - \varphi(\lambda)) \,,$$

*where $\mathcal{D} = \{\lambda \in \mathbb{R} : \forall t, \log \mathbb{E}\left[\exp(\lambda Z_t) | \mathcal{F}_{t-1}\right] \leq \varphi(\lambda) < \infty\}$. Assume that $\mathcal{D}$ contains an open neighborhood of $0$. Let $\varphi_{\star,+}^{-1} : \mathbb{R} \to \mathbb{R}_+$ (resp. $\varphi_{\star,-}^{-1}$) be its reverse map on $\mathbb{R}_+$ (resp. $\mathbb{R}_-$), that is*

$$\varphi_{\star,-}^{-1}(z) := \sup\{x \leq 0 : \varphi_\star(x) > z\} \quad and \quad \varphi_{\star,+}^{-1}(z) := \inf\{x \geq 0 : \varphi_\star(x) > z\} \,.$$

*Let $N_n$ be an integer-valued random variable that is $\mathcal{F}$-measurable and almost surely bounded by $n$. Then, for all $c \in (1, n]$ and $\delta \in (0, 1)$,*

$$\mathbb{P}\left[\frac{1}{N_n} \sum_{t=1}^{N_n} Z_t \geq \varphi_{\star,+}^{-1}\left(\frac{c}{N_n} \log\left(\left\lceil \frac{\log(n)}{\log(c)} \right\rceil \frac{1}{\delta}\right)\right)\right] \leq \delta \,,$$

$$\mathbb{P}\left[\frac{1}{N_n} \sum_{t=1}^{N_n} Z_t \leq \varphi_{\star,-}^{-1}\left(\frac{c}{N_n} \log\left(\left\lceil \frac{\log(n)}{\log(c)} \right\rceil \frac{1}{\delta}\right)\right)\right] \leq \delta \,.$$

*Moreover, if $N$ is a possibly unbounded $\mathbb{N}$-valued random variable that is $\mathcal{F}$-measurable, then for all $c > 1$ and $\delta \in (0, 1)$,*

$$\mathbb{P}\left[\frac{1}{N} \sum_{t=1}^{N} Z_t \geq \varphi_{\star,+}^{-1}\left(\frac{c}{N} \log\left[\frac{\log(N)\log(cN)}{\delta \log^2(c)}\right]\right)\right] \leq \delta \,,$$

$$\mathbb{P}\left[\frac{1}{N} \sum_{t=1}^{N} Z_t \leq \varphi_{\star,-}^{-1}\left(\frac{c}{N} \log\left[\frac{\log(N)\log(cN)}{\delta \log^2(c)}\right]\right)\right] \leq \delta \,.$$

We provide an immediate consequence of this lemma for the case of *sub-Gamma* random variables:

**Corollary 1** *Let $Z = (Z_t)_{t \in \mathbb{N}}$ be a sequence of random variables generated by a predictable process, and $\mathcal{F} = (\mathcal{F}_t)_t$ be its natural filtration. Assume for all $t \in \mathbb{N}$, $|Z_t| \leq b$ and $\mathbb{E}[Z_t^2 | \mathcal{F}_{s-1}] \leq v$ for some positive numbers $v$ and $b$. Let $N_n$ be an integer-valued random variable that is $\mathcal{F}$-measurable and almost surely bounded by $n$. Then, for all $c \in (1, n]$ and $\delta \in (0, 1)$,*

$$\mathbb{P}\left[\frac{1}{N_n} \sum_{t=1}^{N_n} Z_t \geq \sqrt{\frac{2\zeta(n, \delta)v}{N_n}} + \frac{\zeta(n, \delta)b}{3N_n}\right] \leq \delta \,,$$

$$\mathbb{P}\left[\frac{1}{N_n} \sum_{t=1}^{N_n} Z_t \leq -\sqrt{\frac{2\zeta(n, \delta)v}{N_n}} - \frac{\zeta(n, \delta)b}{3N_n}\right] \leq \delta \,,$$

*where $\zeta(n, \delta) := c\log\left(\left\lceil \frac{\log(n)}{\log(c)} \right\rceil \frac{1}{\delta}\right)$, with $c > 1$ being an arbitrary parameter.*

*Proof.* The proof follows by an application of Lemma 2 for sub-Gamma random variables with parameters $(v, b)$. Note that sub-Gamma random variables satisfy $\varphi(\lambda) \leq \frac{\lambda^2 v}{2(1 - b\lambda)}$, for all $\lambda \in (0, 1/b)$; see, e.g., [39, Chapter 2.4]. Therefore,

$$\varphi_{\star,+}^{-1}(z) = \sqrt{2vz} + bz \quad and \quad \varphi_{\star,-}^{-1}(z) = -\sqrt{2vz} - bz \,.$$

Plugging these into the first statements of Lemma 2 completes the proof. $\qquad \square$

As a consequence of this corollary, we present the following lemma:

**Lemma 3 (Bernstein-Markov Concentration)** *Let $(X_t)_{1 \leq t \leq n}$ be generated from a Markov chain defined on a finite state-space $\mathcal{S}$ with transition matrix $P$. Consider the smoothed estimator $\widehat{P}_n$ of $P$ defined as follows: For all $(x, y) \in \mathcal{S}^2$,*

$$\widehat{P}_n(x, y) := \frac{\alpha + \sum_{t=2}^n \mathbb{I}\{X_{t-1} = x, X_t = y\}}{\alpha S + T_{x,n}},$$

*with $\alpha > 0$. Then, for any $\delta \in (0, 1)$, it holds that with probability at least $1 - \delta$, for all $(x, y) \in \mathcal{S}^2$,*

$$|\widehat{P}_n(x, y) - P(x, y)| \leq \sqrt{\left(\frac{T_{x,n}}{T_{x,n} + \alpha S}\right) \frac{2[P(I - P)](x, y)\zeta(n, \delta)}{T_{x,n} + \alpha S}} + \frac{\frac{1}{3}\zeta(n, \delta) + \alpha|1 - SP(x, y)|}{T_{x,n} + \alpha S},$$

*where $\zeta(n, \delta) := c\log\left(\left\lceil \frac{\log(n)}{\log(c)} \right\rceil \frac{2S^2}{\delta}\right)$, with $c > 1$ being an arbitrary parameter.*

*Proof.* The proof uses similar steps as in the one of Lemma 8.3 in [36]. Consider a pair $(x, y) \in \mathcal{S}^2$. We have

$$\widehat{P}_n(x, y) - P(x, y) = \frac{\alpha + \sum_{t=2}^n \mathbb{I}\{X_{t-1} = x, X_t = y\}}{\alpha S + T_{x,n}} - P(x, y)$$

$$= \frac{T_{x,n}}{T_{x,n} + \alpha S}Y_n + \frac{\alpha(1 - SP(x, y))}{T_{x,n} + \alpha S},$$

where $Y_n := \frac{1}{T_{x,n}}\left(\sum_{t=2}^n \mathbb{I}\{X_{t-1} = x, X_t = y\} - T_{x,n}P(x, y)\right)$. Hence,

$$|\widehat{P}_n(x, y) - P(x, y)| \leq \frac{T_{x,n}}{T_{x,n} + \alpha S}|Y_n| + \frac{\alpha|1 - SP(x, y)|}{T_{x,n} + \alpha S}. \tag{3}$$

To control $Y_n$, similarly to the proof of [36, Lemma 8.3], we define the sequence $(Z_t)_{1 \leq t \leq n}$, with $Z_1 := 0$, and

$$Z_t := \mathbb{I}\{X_{t-1} = x\}(\mathbb{I}\{X_t = y\} - P(x, y)), \quad \forall t \geq 2.$$

Note that for all $t$, $Z_t \in [-P(x, y), 1 - P(x, y)]$ almost surely. Moreover, denoting by $(\mathcal{F}_t)_t$ the filtration generated by $(X_t)_{1 \leq t \leq n}$, we observe that $(Z_t)_{1 \leq t \leq n}$ is $\mathcal{F}_{t-1}$-measurable and $\mathbb{E}[Z_t|\mathcal{F}_{t-1}] = 0$. Hence, it is a bounded martingale difference sequence with respect to $(\mathcal{F}_t)_t$, and

$$\mathbb{E}[Z_t^2|\mathcal{F}_{t-1}] = P(x, y)(1 - P(x, y))\mathbb{I}\{X_{t-1} = x\}, \quad \forall t \geq 2.$$

Applying Corollary 1 yields

$$|Y_n| \leq \sqrt{\frac{2P(x, y)(1 - P(x, y))\zeta(n, \delta)}{T_{x,n}}} + \frac{\zeta(n, \delta)}{3T_{x,n}},$$

with probability at least $1 - \frac{\delta}{S^2}$. Plugging the above bound into (3) and taking a union bound complete the proof. $\square$

**Lemma 4 (Empirical Bernstein-Markov Concentration)** *Let $(X_t)_{1 \leq t \leq n}$ be generated from a Markov chain defined on $\mathcal{S}$ with transition matrix $P$. Consider the smoothed estimator $\widehat{P}_n$ of $P$ as defined in Lemma 3. Then, with probability at least $1 - \delta$, for all $(x, y) \in \mathcal{S}^2$,*

$$|\widehat{P}_n(x, y) - P(x, y)| \leq \left(\frac{2T_{x,n}[\widehat{P}_n(I - \widehat{P}_n)](x, y)\zeta}{(T_{x,n} + \alpha S)^2} + c_1\frac{\sqrt{T_{x,n}[\widehat{P}_n(I - \widehat{P}_n)](x, y)}}{(T_{x,n} + \alpha S)^2} + \frac{c_2}{(T_{x,n} + \alpha S)^2}\right)^{1/2},$$

*where $\zeta := \zeta(n, \delta) := c\log\left(\left\lceil \frac{\log(n)}{\log(c)} \right\rceil \frac{2S^2}{\delta}\right)$, where $c > 1$ is an arbitrary parameter, $\zeta' := \frac{1}{3}\zeta + \alpha(S - 1)$, and*

$$c_1 = \sqrt{8\zeta}(2\zeta + \zeta') \quad \text{and} \quad c_2 := \zeta'^2 + 4\zeta(4\zeta + \zeta' + 2\sqrt{\zeta\zeta'}) + \zeta'\sqrt{8\zeta}(5.3\sqrt{\zeta} + \sqrt{2\zeta'}).$$

*Proof.* Fix a pair $(x, y) \in \mathcal{S}^2$. Recall from Lemma 3 that with probability at least $1 - \delta$,

$$|\widehat{P}_n(x,y) - P(x,y)| \leq \sqrt{\frac{2\zeta T_{x,n}[P(I-P)](x,y)}{(T_{x,n} + \alpha S)^2}} + \frac{\zeta'}{T_{x,n} + \alpha S},$$

so that

$$(\widehat{P}_n(x,y) - P(x,y))^2 \leq \frac{2\zeta T_{x,n}[P(I-P)](x,y)}{(T_{x,n} + \alpha S)^2} + \frac{\zeta'^2}{(T_{x,n} + \alpha S)^2} + \sqrt{\frac{8\zeta T_{x,n}[P(I-P)](x,y)}{(T_{x,n} + \alpha S)^2}} \frac{\zeta'}{T_{x,n} + \alpha S}.$$
(4)

Next we derive an upper bound on $[P(I-P)](x,y)$. By Taylor's expansion, we have

$$[P(I-P)](x,y) = [\widehat{P}_n(I - \widehat{P}_n)](x,y) + [(I - 2\widehat{P}_n)(P - \widehat{P}_n)](x,y) - (P - \widehat{P}_n)(x,y)^2$$

$$= [\widehat{P}_n(I - \widehat{P}_n)](x,y) + [(I - \widehat{P}_n - P)(P - \widehat{P}_n)](x,y)$$

$$\leq [\widehat{P}_n(I - \widehat{P}_n)](x,y) + |(I - \widehat{P}_n - P)(x,y)| \left( \sqrt{\frac{2\zeta T_{x,n}[P(I-P)](x,y)}{(T_{x,n} + \alpha S)^2}} + \frac{\zeta'}{T_{x,n} + \alpha S} \right)$$

$$\leq [\widehat{P}_n(I - \widehat{P}_n)](x,y) + \sqrt{\frac{8\zeta T_{x,n}[P(I-P)](x,y)}{(T_{x,n} + \alpha S)^2}} + \frac{2\zeta'}{T_{x,n} + \alpha S}.$$

Using the fact that $a \leq b\sqrt{a} + c$ implies $a \leq b^2 + b\sqrt{c} + c$ for nonnegative numbers $a, b, c$, we get

$$[P(I-P)](x,y) \leq [\widehat{P}_n(I - \widehat{P}_n)](x,y) + \frac{2\zeta'}{T_{x,n} + \alpha S} + \sqrt{\frac{8\zeta T_{x,n}}{(T_{x,n} + \alpha S)^2} \left( [\widehat{P}_n(I - \widehat{P}_n)](x,y) + \frac{2\zeta'}{T_{x,n} + \alpha S} \right)}$$

$$+ \frac{8\zeta T_{x,n}}{(T_{x,n} + \alpha S)^2}$$

$$\leq [\widehat{P}_n(I - \widehat{P}_n)](x,y) + \sqrt{\frac{8\zeta T_{x,n}}{(T_{x,n} + \alpha S)^2}[\widehat{P}_n(I - \widehat{P}_n)](x,y)} + \frac{8\zeta + 2\zeta' + 4\sqrt{\zeta\zeta'}}{T_{x,n} + \alpha S},$$
(5)

where we have used $\sqrt{a + b} \leq \sqrt{a} + \sqrt{b}$ valid for all $a, b \geq 0$. Taking square-root from both sides and using the fact $\sqrt{a + b} \leq \sqrt{a} + \frac{b}{2\sqrt{a}}$ valid for all $a, b > 0$ give

$$\sqrt{[P(I-P)](x,y)} \leq \sqrt{[\widehat{P}_n(I - \widehat{P}_n)](x,y)} + \frac{1}{\sqrt{T_{x,n} + \alpha S}} \left( \sqrt{2\zeta} + \sqrt{8\zeta + 2\zeta' + 4\sqrt{\zeta\zeta'}} \right)$$

$$\leq \sqrt{[\widehat{P}_n(I - \widehat{P}_n)](x,y)} + \frac{5.3\sqrt{\zeta} + \sqrt{2\zeta'}}{\sqrt{T_{x,n} + \alpha S}},$$
(6)

where we have used

$$\sqrt{2\zeta} + \sqrt{8\zeta + 2\zeta' + 4\sqrt{\zeta\zeta'}} \leq \sqrt{2\zeta} + \sqrt{6\zeta + 2(\sqrt{\zeta} + \sqrt{\zeta'})^2} \leq 5.3\sqrt{\zeta} + \sqrt{2\zeta'}.$$

Plugging (5) and (6) into (4), we obtain

$$(\widehat{P}_n(x,y) - P(x,y))^2$$

$$\leq \frac{2\zeta T_{x,n}}{(T_{x,n} + \alpha S)^2} \left( [\widehat{P}_n(I - \widehat{P}_n)](x,y) + \sqrt{\frac{8\zeta T_{x,n}}{(T_{x,n} + \alpha S)^2}[\widehat{P}_n(I - \widehat{P}_n)](x,y)} + \frac{8\zeta + 2\zeta' + 4\sqrt{\zeta\zeta'}}{T_{x,n} + \alpha S} \right)$$

$$+ \frac{\zeta'}{T_{x,n} + \alpha S} \sqrt{\frac{8\zeta T_{x,n}}{(T_{x,n} + \alpha S)^2}} \left( \sqrt{[\widehat{P}_n(I - \widehat{P}_n)](x,y)} + \frac{5.3\sqrt{\zeta} + \sqrt{2\zeta'}}{\sqrt{T_{x,n} + \alpha S}} \right) + \frac{\zeta'^2}{(T_{x,n} + \alpha S)^2}$$

$$\leq \frac{2\zeta T_{x,n}[\widehat{P}_n(I - \widehat{P}_n)](x,y)}{(T_{x,n} + \alpha S)^2} + c_1 \frac{\sqrt{T_{x,n}[\widehat{P}_n(I - \widehat{P}_n)](x,y)}}{(T_{x,n} + \alpha S)^2} + \frac{c_2}{(T_{x,n} + \alpha S)^2},$$

with

$$c_1 := \sqrt{8\zeta}(2\zeta + \zeta') \quad \text{and} \quad c_2 := \zeta'^2 + 4\zeta(4\zeta + \zeta' + 2\sqrt{\zeta\zeta'}) + \zeta'\sqrt{8\zeta}(5.3\sqrt{\zeta} + \sqrt{2\zeta'}),$$

which after taking the square-root from both sides yields the announced result. □

Next we recall a result for the convergence of empirical stationary distributions in an ergodic Markov chain to its stationary distribution:

**Lemma 5 ([20])** *Let $(X_t)_{1 \leq t \leq n}$ be an ergodic and reversible Markov chain defined on $\mathcal{S}$ with stationary distribution $\pi$ and spectral gap $\gamma$. Let $\hat{\pi}_n$ denote the corresponding empirical stationary distribution of the Markov chain. For any $\delta \in (0,1)$, with probability at least $1 - \delta$,*

$$|\hat{\pi}_n(x) - \pi(x)| \leq \sqrt{\frac{8\pi(x)(1-\pi(x))}{\gamma n} \log\left(\frac{S}{\delta}\sqrt{\frac{2}{\min_x \pi(x)}}\right)} + \frac{20}{\gamma n} \log\left(\frac{S}{\delta}\sqrt{\frac{2}{\min_x \pi(x)}}\right), \quad \forall x \in \mathcal{S}.$$

# B    Technical Lemmas

Before providing the proofs of the main theorems, we provide some technical lemmas. We begin with the following definition:

**Definition 2 (Definition of the Event $C$)** *Let $n \geq 1$ and $\delta > 0$. For any $(x,y) \in \mathcal{S}^2$ and $k \in [K]$ define*

$$C_{x,y,k}(n,\delta) := \left\{ \forall t \leq n : |(\widehat{P}_{k,t} - P_k)(x,y)| \leq \sqrt{\frac{2T_{k,x,t}[P_k(I - P_k)](x,y)\beta(n,\delta)}{(T_{k,x,t} + \alpha S)^2}} + \frac{\beta(n,\delta)}{3(T_{k,x,t} + \alpha S)} \right\},$$

*where $\beta(n,\delta) := c\log\left(\left\lceil \frac{\log(n)}{\log(c)} \right\rceil \frac{6KS^2}{\delta}\right)$. Define*

$$C := C(n,\delta) := \cap_{k \in [K]} \cap_{x,y \in \mathcal{S}} C_{x,y,k}(n,\delta).$$

**Lemma 6** *For any $\alpha \leq \frac{1}{3S}$, $n \geq 1$, and $\delta \in (0,1)$, it holds that $\mathbb{P}(C(n,\delta)) \geq 1 - \delta$.*

*Proof.* Let $n \geq 1$ and $\delta > 0$. Define $\zeta(n,\delta) = c\log\left(\left\lceil \frac{\log(n)}{\log(c)} \right\rceil \frac{2KS^2}{\delta}\right)$, and note that $\beta(n,\delta) = \zeta(n,\delta) + c\log(3)$. Applying Lemma 3, we obtain for any chain $k$:

$$|(\widehat{P}_{k,t} - P_k)(x,y)| \leq \sqrt{\frac{2T_{k,x,t}[P_k(I - P_k)](x,y)\zeta(n,\delta)}{(T_{k,x,t} + \alpha S)^2}} + \frac{\frac{1}{3}\zeta(n,\delta) + \alpha(S-1)}{T_{k,t,x} + \alpha S},$$

for all $(x,y)$, and uniformly for all $t \leq n$, with probability at least $1 - \frac{\delta}{K}$. With the choice of $\alpha = \frac{1}{3S}$, and noting that $\beta(n,\delta) \geq \zeta(n,\delta)$ and

$$\frac{\zeta(n,\delta)}{3} + \alpha(S-1) \leq \frac{\zeta(n,\delta)}{3} + \frac{S-1}{3S} \leq \frac{\beta(n,\delta)}{3},$$

we obtain $\mathbb{P}(\cap_{x,y \in \mathcal{S}} C_{x,y,k}(n,\delta)) \geq 1 - \delta/K$ for all $k$. Finally, using a union bound concludes the proof. □

In the following lemma, we provide an upper bound on the loss $L_{k,n}$, which is valid for all $n$. To simplify the notation, in the following results we let $\beta = \beta(n,\delta)$.

**Lemma 7 (Upper Bound on the Loss)** *Assume that the event $C$ holds. Then, for any budget $n > 1$ and chain $k$,*

$$L_{k,n} \leq \frac{2\beta}{T_{k,n}} \sum_x G_k(x)\mathbb{I}\{T_{k,x,n} > 0\} + \frac{2\sqrt{2}}{3}\frac{\beta^{3/2}\sqrt{S}}{T_{k,n}} \sum_x \frac{T_{k,x,n}^{3/2}\sqrt{G_k(x)}}{(T_{k,x,n} + \alpha S)^2} + \frac{S\beta^2}{9T_{k,n}} \sum_x \frac{T_{k,x,n}}{(T_{k,x,n} + \alpha S)^2}.$$

*Proof.* Let $n > 1$ and consider a chain $k$. To simplify the notation, we omit the dependence of various quantities on $k$ (hence $T_{x,n} := T_{k,x,n}$, $T_n := T_{k,n}$, and so on). On the event $C$, we have

$$|\widehat{P}_n(x,y) - P(x,y)| \leq \sqrt{\left(\frac{T_{x,n}}{T_{x,n} + \alpha S}\right)\frac{2[P(I-P)](x,y)\beta}{T_{x,n} + \alpha S}} + \frac{\beta}{3(T_{x,n} + \alpha S)},$$

so that

$$(\widehat{P}_n(x,y) - P(x,y))^2 \leq \frac{2\beta[P(I-P)](x,y)}{T_{x,n} + \alpha S} + \frac{\frac{1}{9}\beta^2}{(T_{x,n} + \alpha S)^2} + \frac{\frac{2\sqrt{2}}{3}\beta^{3/2}}{(T_{x,n} + \alpha S)^2}\sqrt{T_{x,n}[P(I-P)](x,y)}.$$

Hence, we obtain the announced upper bound on the loss:

$$L_n = \sum_x \hat{\pi}_n(x) \sum_y (\widehat{P}_n(x,y) - P(x,y))^2$$

$$\leq \frac{2\beta}{T_n}\sum_x \frac{T_{x,n}G(x)}{T_{x,n} + \alpha S} + \frac{2\sqrt{2}}{3}\frac{\beta^{3/2}}{T_n}\sum_x \frac{T_{x,n}^{3/2}}{(T_{x,n} + \alpha S)^2}\sum_y \sqrt{[P(I-P)](x,y)} + \frac{S\beta^2}{9T_n}\sum_x \frac{T_{x,n}}{(T_{x,n} + \alpha S)^2}$$

$$\leq \frac{2\beta}{T_n}\sum_x G(x)\mathbb{I}\{T_{x,n} > 0\} + \frac{2\sqrt{2}}{3}\frac{\beta^{3/2}\sqrt{S}}{T_n}\sum_x \frac{T_{x,n}^{3/2}\sqrt{G(x)}}{(T_{x,n} + \alpha S)^2} + \frac{S\beta^2}{9T_n}\sum_x \frac{T_{x,n}}{(T_{x,n} + \alpha S)^2},$$

where the last step follows from the Cauchy-Schwarz inequality. $\qquad\square$

The following lemma presents bounds on the index $b_{k,\cdot}$ on the event $C$ (defined in Definition 2):

**Lemma 8 (Bounds on the Index)** *Consider a chain $k$, and assume that the event $C$ holds. Then, for any time $t$,*

$$b_{k,t+1} \leq \frac{2\beta}{T_{k,t}}\sum_x G_k(x)\mathbb{I}\{T_{k,x,t} > 0\} + \frac{13\beta^{3/2}\sqrt{S}}{T_{k,t}}\sum_x \sqrt{\frac{G_k(x)\mathbb{I}\{T_{k,x,t} > 0\}}{T_{k,x,t} + \alpha S}} + \frac{39\beta^2 S}{T_{k,t}}\sum_x \frac{\mathbb{I}\{T_{k,x,t} > 0\}}{T_{k,x,t} + \alpha S},$$

$$b_{k,t+1} \geq \frac{2\beta}{T_{k,t}}\sum_x G_k(x)\mathbb{I}\{T_{k,x,t} > 0\}.$$

*Proof.* Fix a chain $k$ and time $t$. To ease the notation, let us omit the dependence of various quantities on $k$ throughout. We first recall the definition of the index $b_{t+1}$:

$$b_{t+1} = \frac{2\beta}{T_t}\sum_x \widehat{G}_t(x)\mathbb{I}\{T_{x,t} > 0\} + \frac{c_1}{T_t}\sum_x \frac{T_{x,t}^{3/2}}{(T_{x,t} + \alpha S)^2}\sum_y \sqrt{[\widehat{P}_t(I - \widehat{P}_t)](x,y)} + \frac{c_2 S}{T_t}\sum_x \frac{\mathbb{I}\{T_{x,t} > 0\}}{T_{x,t} + \alpha S},$$

where $c_1 = 6.6\beta^{3/2}$ and $c_2 = 28\beta^2$.

To derive an upper bound on $b_t$, we first find an upper bound on $[\widehat{P}_t(I - \widehat{P}_t)](x,y)$ as follows. First, using Taylor's expansion, we have

$$[\widehat{P}_t(I - \widehat{P}_t)](x,y) = [P(I - P)](x,y) + [(I - 2P)(\widehat{P}_t - P)](x,y) - (\widehat{P}_t - P)(x,y)^2$$

$$= [P(I - P)](x,y) + [(I - P - \widehat{P}_t)(\widehat{P}_t - P)](x,y)$$

$$\leq [P(I - P)](x,y) + \sqrt{\frac{8\beta[P(I-P)](x,y)}{T_{x,t} + \alpha S}} + \frac{2\beta}{3(T_{x,t} + \alpha S)} \qquad (7)$$

$$\leq \left(\sqrt{[P(I-P)](x,y)} + \sqrt{\frac{2\beta}{T_{x,t} + \alpha S}}\right)^2,$$

where (7) follows from the definition of $C$. Hence,

$$\sqrt{[\widehat{P}_t(I - \widehat{P}_t)](x,y)} \leq \sqrt{[P(I-P)](x,y)} + \sqrt{\frac{2\beta}{T_{x,t} + \alpha S}}. \qquad (8)$$

Using (7) and (8), we obtain the following upper bound on $b_t$, on the event $C$:

$$b_{t+1} \leq \frac{2\beta}{T_t} \sum_x \sum_y \left( [P(I-P)](x,y) + \sqrt{\frac{8\beta[P(I-P)](x,y)}{T_{x,t}+\alpha S}} + \frac{2\beta}{3(T_{x,t}+\alpha S)} \right) \mathbb{I}\{T_{x,t} > 0\}$$

$$+ \frac{6.6\beta^{3/2}}{T_t} \sum_x \frac{T_{x,t}^{3/2}}{(T_{x,t}+\alpha S)^2} \sum_y \left( \sqrt{[P(I-P)](x,y)} + \sqrt{\frac{2\beta}{T_{x,t}+\alpha S}} \right) + \frac{28\beta^2 S}{T_t} \sum_x \frac{\mathbb{I}\{T_{x,t} > 0\}}{T_{x,t}+\alpha S}$$

$$\leq \frac{2\beta}{T_t} \sum_x G(x)\mathbb{I}\{T_{x,t} > 0\} + \frac{13\beta^{3/2}\sqrt{S}}{T_t} \sum_x \sqrt{\frac{G(x)\mathbb{I}\{T_{x,t} > 0\}}{T_{x,t}+\alpha S}} + \frac{39\beta^2 S}{T_t} \sum_x \frac{\mathbb{I}\{T_{x,t} > 0\}}{T_{x,t}+\alpha S}.$$

To prove the lower bound on the index, we recall from the proof of Lemma 4 (see (5) with the choices $\zeta = \beta$ and $\zeta' = \frac{\beta}{3}$) that

$$[\widehat{P}_t(I - \widehat{P}_t)](x,y) \geq [P(I-P)](x,y) - \sqrt{\frac{8\beta T_{x,t}}{(T_{x,t}+\alpha S)^2}[\widehat{P}_t(I - \widehat{P}_t)](x,y)} - \frac{12}{T_{x,t}+\alpha S}.$$

Putting this together with the definition of $b_{t+1}$ leads to $b_{t+1} \geq \frac{2\beta}{T_t} \sum_x G(x)\mathbb{I}\{T_{x,t} > 0\}$, and thus completes the proof. $\qquad\square$

## C  Proof of Theorem 1

Consider a chain $k$ and assume that the event $C$ (defined in Definition 2) holds. Applying Lemma 7, we obtain

$$L_{k,n} \leq \frac{2\beta}{T_{k,n}} \sum_{x:T_{k,x,n}>0} G_k(x) + \frac{2\sqrt{2}}{3}\frac{\beta^{3/2}\sqrt{S}}{T_{k,n}} \sum_x \frac{T_{k,x,n}^{3/2}\sqrt{G_k(x)}}{(T_{k,x,n}+\alpha S)^2} + \frac{S\beta^2}{9T_{k,n}} \sum_x \frac{T_{k,x,n}}{(T_{k,x,n}+\alpha S)^2}$$

$$\leq \frac{2\beta}{T_{k,n}} \sum_{x:T_{k,x,n}>0} G_k(x) + \frac{2\sqrt{2}}{3}\frac{\beta^{3/2}}{T_{k,n}} \sqrt{S \sum_{x:T_{k,x,n}>0} G_k(x)} \sqrt{\sum_{x:T_{k,x,n}>0} \frac{1}{T_{k,x,n}+\alpha S}}$$

$$+ \frac{S\beta^2}{9T_{k,n}} \sum_{x:T_{k,x,n}>0} \frac{1}{T_{k,x,n}+\alpha S}$$

$$= \left( \sqrt{\frac{2\beta}{T_{k,n}} \sum_{x:T_{k,x,n}>0} G_k(x)} + \sqrt{\frac{S\beta^2}{9T_{k,n}} \sum_{x:T_{k,x,n}>0} \frac{1}{T_{k,x,n}+\alpha S}} \right)^2,$$

where we have used Cauchy-Schwarz in the second inequality. Introducing

$$A_{1,k} := \frac{S\beta^2}{9T_{k,n}} \sum_{x:T_{k,x,n}>0} \frac{1}{T_{k,x,n}+\alpha S} \quad \text{and} \quad A_{2,k} := \frac{2\beta}{T_{k,n}} \sum_{x:T_{k,x,n}>0} G_k(x),$$

we provide upper bounds on $A_{1,k}$ and $A_{2,k}$ in the following lemmas:

**Lemma 9** *On the event $C$, it holds for any chain $k$ and any $n$:*

$$A_{1,k} \leq \frac{0.175KS^2\beta^2}{n-K}.$$

**Lemma 10** *Assume that the event $C$ holds. Then for any chain $k$ and $n$:*

$$A_{2,k} \leq \frac{272KS^2\beta^2}{n-2K} + \frac{518K^2S^2\beta^2}{(n-2K)^2}.$$

Applying Lemmas 9 and 10 gives

$$L_{k,n} \le (\sqrt{A_{1,k}} + \sqrt{A_{2,k}})^2 \le \frac{KS^2\beta^2}{n-2K}\left(\sqrt{272 + \frac{518K}{n-2K}} + \sqrt{0.175}\right)^2$$

$$\le KS^2\beta^2\left(\frac{287}{n-2K} + \frac{532K}{(n-2K)^2}\right),$$

where we have used

$$\left(\sqrt{272 + \frac{518K}{n-2K}} + \sqrt{0.175}\right)^2 \le 273 + \frac{518K}{n-2K} + 2\sqrt{0.175}\sqrt{272 + \frac{518K}{n-2K}} \le 287 + \frac{532K}{n-2K}.$$

Finally, using the inequality $(n-2K)^{-1} \le n^{-1} + 4Kn^{-2}$ valid for $n \ge 4K$, and noting that the event $C$ holds with probability higher than $1 - \delta$, we get the desired bound on the loss. ☐

## C.1 Proof of Lemma 9

Assume that $C$ holds. We claim that there exists a chain $j$ such that $T_{j,n} \ge \frac{n}{K}$. Indeed, if for all $j$, $T_{j,n} < \frac{n}{K}$, then $\sum_{j=1}^{K} T_{j,n} < n$, which is a contradiction. Hence the claim holds.

Now, consider a chain $j$ such that $T_{j,n} \ge \frac{n}{K}$. Let $t + 1 \le n$ be the last time that it has been sampled. Hence, $T_{j,t+1} = T_{j,n}$ and $T_{j,t} = T_{j,n} - 1 \ge \frac{n}{K} - 1$. Applying Lemma 8 for chain $j$, it follows that on the event $C$,

$$b_{j,t+1} \le \frac{2\beta}{T_{j,t}}\sum_{x:T_{j,x,t}>0} G_j(x) + \frac{13\beta^{3/2}\sqrt{S}}{T_{j,t}}\sum_{x:T_{j,x,t}>0}\sqrt{\frac{G_j(x)}{T_{j,x,t}+\alpha S}} + \frac{39\beta^2 S}{T_{j,t}}\sum_{x:T_{j,x,t}>0}\frac{1}{T_{j,x,t}+\alpha S}$$

$$\le \frac{2\beta}{T_{j,t}}\sum_{x:T_{j,x,t}>0} G_j(x) + \frac{13\beta^{3/2}\sqrt{S}}{T_{j,t}}\sqrt{\sum_{x:T_{j,x,t}>0} G_j(x)}\sqrt{\sum_{x:T_{j,x,t}>0}\frac{1}{T_{j,x,t}+\alpha S}}$$

$$+ \frac{39\beta^2 S}{T_{j,t}}\sum_{x:T_{j,x,t}>0}\frac{1}{T_{j,x,t}+\alpha S}$$

$$\le \frac{K}{n-K}\left(2\beta\sum_x G_j(x) + 13\beta^{3/2}\sqrt{S\sum_x G_j(x)}\sqrt{\sum_x \frac{1}{1+\alpha S}} + 39\beta^2 S\sum_x \frac{1}{1+\alpha S}\right)$$

$$\le \frac{K}{n-K}\left(2\beta\sum_x G_j(x) + 12S\beta^{3/2}\sqrt{\sum_x G_j(x)} + 30\beta^2 S^2\right),$$

where the last step uses $\alpha = \frac{1}{3S}$. Noting that $\sum_x G_j(x) \le S - 1$, we get

$$b_{j,t+1} \le \frac{K}{n-K}\left(2\beta(S-1) + 12S\beta^{3/2}\sqrt{S-1} + 30\beta^2 S^2\right) \le \frac{44KS^2\beta^2}{n-K},$$

where we have used that $S \ge 2$. Note that for any chain $i$, by the definition of $b_{i,t+1}$,

$$b_{i,t+1} \ge \frac{28\beta^2 S}{T_{i,t}}\sum_x \frac{\mathbb{I}\{T_{i,x,t} > 0\}}{T_{i,x,t}+\alpha S} \ge \frac{28\beta^2 S}{T_{i,n}}\sum_x \frac{\mathbb{I}\{T_{i,x,n} > 0\}}{T_{i,x,n}+\alpha S}.$$

Furthermore, since $j$ is played at time $t$, it holds that for any chain $i \neq j$, $b_{i,t+1} \le b_{j,t+1}$, so that for any chain $i$,

$$b_{j,t+1} \ge b_{i,t+1} \ge \frac{28\beta^2 S}{T_{i,n}}\sum_x \frac{\mathbb{I}\{T_{i,x,n} > 0\}}{T_{i,x,n}+\alpha S}.$$

Thus, combining this with the upper bound on $b_{j,t+1}$ leads to the desired results. ☐

## C.2 Proof of Lemma 10

The proof borrows some ideas from the proof of Lemma 1 in [2]. Consider a chain $j$ that is sampled at least once after initialization, and let $t + 1(> 2K)$ be the last time it was sampled. Hence, $T_{j,t} = T_{j,n} - 1$ and $T_{j,t+1} = T_{j,n}$. Moreover, let $X_{t+1}$ be the observed state of $j$ at $t + 1$. Then, $T_{j,X_{t+1},t} = T_{j,X_{t+1},n} - 1$ and $T_{j,X_{t+1},t+1} = T_{j,X_{t+1},n}$, whereas for all $x \neq X_{t+1}$, $T_{j,x,t} = T_{j,x,t+1} = T_{j,x,n}$. We thus have, $T_{j,x,t} \geq T_{j,x,n} - 1$ for all $x \in \mathcal{S}$.

By the design of the algorithm, for any chain $k$, $b_{k,t+1} \leq b_{j,t+1}$. Using Lemma 8 yields

$$
\frac{2\beta}{T_{k,n}} \sum_{x:T_{k,x,n}>0} G_k(x) \leq \frac{2\beta}{T_{j,n}-1} \sum_{x:T_{j,x,t}>0} G_j(x) + \frac{13\beta^{3/2}\sqrt{S}}{T_{j,n}-1} \sum_{x:T_{j,x,t}>0} \sqrt{\frac{G_j(x)}{T_{j,x,t}+\alpha S}}
$$

$$
+ \frac{39\beta^2 S}{T_{j,n}-1} \sum_{x:T_{j,x,t}>0} \frac{1}{T_{j,x,t}+\alpha S}
$$

$$
\leq \frac{2\beta}{T_{j,n}-1} \sum_{x:T_{j,x,n}>0} G_j(x) + \frac{26\beta^{3/2}\sqrt{S}}{T_{j,n}-1} \sum_{x:T_{j,x,n}>0} \sqrt{\frac{G_j(x)}{T_{j,x,n}+\alpha S}}
$$

$$
+ \frac{156\beta^2 S}{T_{j,n}-1} \sum_{x:T_{j,x,n}>0} \frac{1}{T_{j,x,n}+\alpha S} \, ,
$$

where in the second inequality, we have used that for $\alpha = \frac{1}{3S}$ and $T_{j,x,n} \geq 1$,

$$
T_{j,x,t} + \alpha S \geq T_{j,x,n} - 1 + \alpha S \geq \frac{T_{j,x,n}+\alpha S}{4} \, .
$$

The above holds for any chain $k$, and any chain $j$ that is sampled at least once after initialization (hence, $T_{j,n} > 2$). Summing over such choices of $j$ gives

$$
\frac{2\beta}{T_{k,n}} \sum_{x:T_{k,x,n}>0} G_k(x) \sum_{j:T_{j,n}>2} (T_{j,n}-1)
$$

$$
\leq 2\beta \sum_j \sum_x G_j(x) + 26\beta^{3/2}\sqrt{S} \sum_j \sum_{x:T_{j,x,n}>0} \sqrt{\frac{G_j(x)}{T_{j,x,n}+\alpha S}} + 156\beta^2 S \sum_j \sum_{x:T_{j,x,n}>0} \frac{1}{T_{j,x,n}+\alpha S}
$$

$$
\leq 2\beta\Lambda + 26\beta^{3/2}\sqrt{S\Lambda}\sqrt{\sum_j \sum_{x:T_{j,x,n}>0} \frac{1}{T_{j,x,n}+\alpha S}} + 156\beta^2 S \sum_j \sum_{x:T_{j,x,n}>0} \frac{1}{T_{j,x,n}+\alpha S} \, ,
$$

where we have used Cauchy-Schwarz in the last inequality, and that $\sum_j \sum_x G_j(x) = \Lambda$. Noting that $\sum_{j:T_{j,n}>2}(T_{j,n}-1) \geq n - 2K$ yields

$$
\frac{2\beta}{T_{k,n}} \sum_{x:T_{k,x,n}>0} G_k(x) \leq \frac{2\beta\Lambda}{n-2K} + \frac{26\beta^{3/2}\sqrt{\Lambda}}{n-2K}\sqrt{\sum_j \sum_{x:T_{j,x,n}>0} \frac{S}{T_{j,x,n}+\alpha S}}
$$

$$
+ \frac{156\beta^2}{n-2K} \sum_j \sum_{x:T_{j,x,n}>0} \frac{S}{T_{j,x,n}+\alpha S} \, .
$$

By Lemma 9, $\sum_{x:T_{j,x,n}>0} \frac{S}{T_{j,x,n}+\alpha S} \leq \frac{1.58KS^2}{n-K}T_{j,n}$ for any chain $j$. Thus,

$$
\frac{2\beta}{T_{k,n}} \sum_{x:T_{j,x,n}>0} G_k(x) \leq \frac{2\beta\Lambda}{n-2K} + \frac{33\beta^{3/2}\sqrt{\Lambda}}{n-2K}\sqrt{\frac{KS^2}{n-K}\sum_j T_{j,n}} + \frac{247KS^2\beta^2}{(n-2K)(n-K)}\sum_j T_{j,n}
$$

$$
\leq \frac{2\beta KS}{n-2K} + \frac{33\beta^{3/2}KS^{3/2}}{n-2K}\sqrt{\frac{n}{n-2K}} + \frac{247KS^2\beta^2 n}{(n-2K)^2}
$$

$$
\leq \frac{272KS^2\beta^2}{n-2K} + \frac{518K^2S^2\beta^2}{(n-2K)^2} \, ,
$$

where we have used $\sum_j T_{j,n} = n$, $\Lambda \leq KS$, $S \geq 2$, and $\sqrt{\frac{n}{n-2K}} \leq 1 + \frac{K}{n-2K}$. $\qquad\square$

# D  Proof of Theorem 2

Let $\delta \in (0,1)$ and $n \geq n_{\mathrm{cutoff}}(\delta)$. To control the loss in this case, we first state the following result for the concentration of empirical state distribution $\hat{\pi}_{k,n}$.

**Lemma 11 (Concentration of Empirical State Distributions)** *Assume that $C$ holds, and define*
$$E := \cap_{k\in[K]} \cap_{x\in\mathcal{S}} \left\{ \hat{\pi}_{k,n}(x)^{-1} \leq 2\pi_k(x)^{-1} \right\}.$$
*For any $\delta \in (0,1)$, if $n \geq n_{\mathrm{cutoff}}(\delta)$, then $\mathbb{P}(E) \geq 1 - \delta$.*

Recalling that $\hat{\pi}_{k,n}(x) = \frac{T_{k,x,n}}{T_{k,n}}$ for all $x \in \mathcal{S}$, and $H_k = \sum_{x\in\mathcal{S}} \pi_k(x)^{-1}$, on the events $C$ and $E$, we have by Lemma 7 and Lemma 11,

$$L_{k,n} \leq \frac{2\beta}{T_{k,n}} \sum_x G_k(x) + \frac{2\sqrt{2}}{3} \frac{\beta^{3/2}\sqrt{S}}{T_{k,n}^{3/2}} \sum_x \sqrt{\frac{G_k(x)}{\hat{\pi}_{k,n}(x)}} + \frac{S\beta^2}{9T_{k,n}^2} \sum_x \frac{1}{\hat{\pi}_{k,n}(x)}$$

$$\leq \frac{2\beta}{T_{k,n}} \sum_x G_k(x) + \frac{4\beta^{3/2}\sqrt{S}}{3T_{k,n}^{3/2}} \sum_x \sqrt{\frac{G_k(x)}{\pi_k(x)}} + \frac{2SH_k\beta^2}{9T_{k,n}^2}$$

$$\leq \frac{2\beta}{T_{k,n}} \sum_x G_k(x) + \frac{4\beta^{3/2}}{3T_{k,n}^{3/2}} \sqrt{SH_k \sum_x G_k(x)} + \frac{2SH_k\beta^2}{9T_{k,n}^2},$$

where the last step follows from the Cauchy-Schwarz inequality.

To control the right-hand side of the above, we first provide an upper bound on $\frac{2\beta}{T_{k,n}} \sum_x G_k(x)$ assuming that $C$ and $E$ occur:

**Lemma 12** *Assume that the events $C$ and $E$ hold. Then, for any chain $k$ and $n \geq n_{\mathrm{cutoff}}$, it holds that*
$$\frac{2\beta}{T_{k,n}} \sum_x G_k(x) \leq \frac{A_1}{n} + \frac{A_2}{n^{3/2}} + \frac{A_3}{n^2} + \widetilde{\mathcal{O}}(n^{-5/2}),$$
*where* $A_1 = 2\beta\Lambda, \quad A_2 = 150\beta^{3/2}K\sqrt{S\Lambda H_{\max}}, \quad A_3 = \frac{3912KSH_{\max}\beta^2}{\eta_{\min}}$.

Applying Lemma 12, and noting $\mathbb{P}(C) \geq 1 - \delta$ (see Lemma 6) and $\mathbb{P}(E) \geq 1 - \delta$, we obtain the following bound on $L_{k,n}$, which holds with probability greater than $1 - 2\delta$:

$$L_{k,n} \leq \frac{2\beta}{T_{k,n}} \sum_x G_k(x) + \frac{4\beta^{3/2}}{3T_{k,n}^{3/2}} \sqrt{SH_k \sum_x G_k(x)} + \frac{2SH_k\beta^2}{9T_{k,n}^2}$$

$$\leq \frac{2\beta}{T_{k,n}} \sum_x G_k(x) + \left(\frac{2\beta}{T_{k,n}} \sum_x G_k(x)\right)^{3/2} \frac{0.48\sqrt{SH_k}}{\sum_x G_k(x)} + \frac{2SH_k}{9(\sum_x G_k(x))^2}\left(\frac{\beta}{T_{k,n}} \sum_x G_k(x)\right)^2$$

$$\leq \frac{A_1}{n} + \frac{A_2}{n^{3/2}} + \frac{A_3}{n^2} + \left(\frac{A_1}{n} + \frac{A_2}{n^{3/2}} + \frac{A_3}{n^2}\right)^{3/2} \frac{0.48\sqrt{SH_k}}{\sum_x G_k(x)}$$

$$+ \frac{2SH_k}{9(\sum_x G_k(x))^2}\left(\frac{A_1}{n} + \frac{A_2}{n^{3/2}} + \frac{A_3}{n^2}\right)^2 + \widetilde{\mathcal{O}}(n^{-5/2})$$

$$\overset{(a)}{\leq} \frac{A_1}{n} + \frac{A_2}{n^{3/2}} + \frac{A_3}{n^2} + \frac{0.84\sqrt{SH_k}}{\sum_x G_k(x)}\left(\frac{A_1^{3/2}}{n^{3/2}} + \frac{A_2^{3/2}}{n^{9/4}} + \frac{A_3^{3/2}}{n^3}\right)$$

$$+ \frac{4SH_k}{9(\sum_x G_k(x))^2}\left(\frac{A_1^2}{n^2} + \frac{A_2^2}{n^3} + \frac{A_3^2}{n^4}\right) + \widetilde{\mathcal{O}}(n^{-5/2})$$

$$\leq \frac{A_1}{n} + \frac{1}{n^{3/2}}\left(A_2 + \frac{0.84\sqrt{SH_k}}{\sum_x G_k(x)}A_1^{3/2}\right) + \widetilde{\mathcal{O}}(n^{-2})$$

$$\leq \frac{2\beta\Lambda}{n} + \frac{\beta^{3/2}}{n^{3/2}}\left(150K\sqrt{S\Lambda H_{\max}} + \frac{2.4\sqrt{SH_k\Lambda}}{\eta_k}\right) + \widetilde{\mathcal{O}}(n^{-2}),$$

where (a) follows from the fact that for positive numbers $a_i, i = 1, \ldots, m$, we have by Jensen's inequality,

$$\left(\frac{1}{m} \sum_{i=1}^{m} a_i\right)^{3/2} \leq \frac{1}{m} \sum_{i=1}^{m} a_i^{3/2} .$$

so that $(\sum_{i=1}^{m} a_i)^{3/2} \leq \sqrt{m} \sum_{i=1}^{m} a_i^{3/2}$. Finally, taking the maximum over $k$ completes the proof. $\square$

### D.1 Proof of Lemma 11

By Lemma 5, we have for all chains $k$ and all $x \in \mathcal{S}$,

$$|\hat{\pi}_{k,n}(x) - \pi_k(x)| \leq \xi_{k,x,n} := \sqrt{\frac{8\pi_k(x)\varepsilon_k}{T_{k,n}}} + \frac{20\varepsilon_k}{T_{k,n}} ,$$

with probability at least $1-\delta$, where $\varepsilon_k := \frac{1}{\gamma_k} \log\left(\frac{KS}{\delta}\sqrt{\frac{2}{\pi_k}}\right)$. It is easy to verify that if $T_{k,n} \geq \frac{96\varepsilon_k}{\pi_k}$, then $\xi_{k,x,n} \leq \pi_k(x)/2$, so that $\hat{\pi}_{k,n}(x) \geq \pi(x)/2$. Hence, for all $k$ and all $x$, $\hat{\pi}_{k,n}(x)^{-1} \leq 2\pi_k(x)^{-1}$ with probability at least $1 - \delta$.

It remains to show that if $n \geq n_{\text{cutoff}}$, we have $T_{k,n} \geq \frac{96\varepsilon_k}{\pi_k}$. Assuming that $C$ occurs, as a consequence of Lemma 9, one has

$$\frac{S\beta^2}{9T_{k,n}} \sum_{x:T_{k,x,n}>0} \frac{1}{T_{k,x,n} + \alpha S} \leq \frac{0.175 K S^2 \beta^2}{n - K} ,$$

Using the trivial bound $T_{k,x,n} \leq T_{k,n}$, it follows that

$$\frac{S^2\beta^2}{9T_{k,n}(T_{k,n} + \alpha S)} \leq \frac{0.175 K S^2 \beta^2}{n - K} ,$$

so that

$$T_{k,n} \geq \sqrt{\frac{n - K}{1.575 K}} - \frac{1}{3} \geq 0.56\sqrt{\frac{n}{K}} - \frac{1}{3} \geq 0.327\sqrt{\frac{n}{K}} .$$

Putting together, we deduce that if $n$ satisfies $0.327\sqrt{\frac{n}{K}} \geq \frac{96\varepsilon_k}{\pi_k}$, we have $T_{k,n} \geq \frac{96\varepsilon_k}{\pi_k}$, and the lemma follows.

Moreover, when the chain $k$ is non-reversible, we may use [20, Theorem 3.4] (instead of Lemma 9), and follow the exact same lines as above to deduce that if $n \geq K \max_k \left(\frac{300}{\gamma_{\text{ps},k}\pi_k} \log\left(\frac{2KS}{\delta}\sqrt{\pi_k^{-1}}\right)\right)^2$, the assertion of the lemma follows. $\square$

### D.2 Proof of Lemma 12

The proof borrows some ideas from the proof of Lemma 1 in [2]. Consider a chain $j$ that is sampled at least once after initialization, and let $t + 1(> 2K)$ be the last time it was sampled. Hence, $T_{j,t} = T_{j,n} - 1$ and $T_{j,t+1} = T_{j,n}$. Moreover, let $X_{t+1}$ be the observed state of $j$ at $t + 1$. Then, $T_{j,X_{t+1},t} = T_{j,X_{t+1},n} - 1$ and $T_{j,X_{t+1},t+1} = T_{j,X_{t+1},n}$, whereas for all $x \neq X_{t+1}$, $T_{j,x,t} = T_{j,x,t+1} = T_{j,x,n}$. We thus have, $T_{j,x,t} \geq T_{j,x,n} - 1$ for all $x \in \mathcal{S}$.

By the design of the algorithm, for any chain $k$, $b_{k,t+1} \leq b_{j,t+1}$. Applying Lemma 8 gives

$$\frac{2\beta}{T_{k,t}} \sum_x G_k(x) \leq \frac{2\beta}{T_{j,t}} \sum_x G_j(x) + \frac{c_3\sqrt{S}}{T_{j,t}} \sum_x \sqrt{\frac{G_j(x)}{T_{j,x,t} + \alpha S}} + \frac{c_4 S}{T_{j,t}} \sum_x \frac{1}{T_{j,x,t} + \alpha S}$$

$$\leq \frac{2\beta}{T_{j,n} - 1} \sum_x G_j(x) + \frac{2c_3\sqrt{S}}{T_{j,n} - 1} \sum_x \sqrt{\frac{G_j(x)}{T_{j,x,n} + \alpha S}} + \frac{4c_4 S}{T_{j,n} - 1} \sum_x \frac{1}{T_{j,x,n} + \alpha S} ,$$

where $c_3 = 13\beta^{3/2}$ and $c_4 = 39\beta^2$, and where where in the second line we have used that for $\alpha = \frac{1}{3S}$ and $T_{j,x,n} \geq 1$

$$T_{j,x,t} + \alpha S \geq T_{j,x,n} - 1 + \alpha S \geq \frac{T_{j,x,n} + \alpha S}{4} .$$

Now, applying Lemma 11 and using $T_{k,t} \leq T_{k,n}$ yield

$$\frac{2\beta}{T_{k,n}} \sum_x G_k(x) \leq \frac{2\beta}{T_{j,n}-1} \sum_x G_j(x) + \frac{c_3\sqrt{8S}}{(T_{j,n}-1)T_{j,n}^{1/2}} \sum_x \sqrt{\frac{G_j(x)}{\pi_j(x)}} + \frac{8c_4 S}{(T_{j,n}-1)T_{j,n}} \sum_x \frac{1}{\pi_j(x)}$$

$$\leq \frac{1}{T_{j,n}-1} \left( 2\beta \sum_x G_j(x) + \frac{\sqrt{8}c_3}{T_{j,n}^{1/2}} \sqrt{SH_j \sum_x G_j(x)} + \frac{8c_4 SH_j}{T_{j,n}} \right).$$

Note that the above relation is valid for any $k$, and any $j$ that is sampled after initialization (i.e., $T_{j,n} > 2$). Summing over such choices of $j$ gives

$$\sum_{j:T_{j,n}>2} \frac{2\beta \sum_x G_k(x)}{T_{k,n}} (T_{j,n}-1) \leq \sum_{j:T_{j,n}>2} \left( 2\beta \sum_x G_j(x) + \frac{c_3\sqrt{8SH_j \sum_x G_j(x)}}{T_{j,n}^{1/2}} + \frac{8c_4 SH_j}{T_{j,n}} \right).$$

Noting that $\sum_{j:T_{j,n}>2}(T_{j,n}-1) \geq n - 2K$, we have

$$\frac{2\beta}{T_{k,n}} \sum_x G_k(x) \leq \frac{1}{n-2K} \sum_j \left( 2\beta \sum_x G_j(x) + \frac{c_3\sqrt{8SH_j \sum_x G_j(x)}}{T_{j,n}^{1/2}} + \frac{8c_4 SH_j}{T_{j,n}} \right)$$

$$\leq \frac{2\beta\Lambda}{n-2K} + \frac{c_3\sqrt{8S}}{n-2K} \sum_j \sqrt{H_j \frac{\sum_x G_j(x)}{T_{j,n}}} + \frac{8c_4 S}{n-2K} \sum_j \frac{H_j}{T_{j,n}}. \qquad (9)$$

To carefully control the right-hand side of the last inequality, we use the following lemma:

**Lemma 13** *Under the same assumptions of Lemma 12, we have for any chain $j$,*

$$\frac{\beta}{T_{j,n}} \leq \frac{\beta}{\eta_{\min}(n-2K)} + \frac{c_3}{\eta_{\min}^2(n-2K)^{3/2}} \sqrt{2SH_{\max}/\Lambda} + \frac{4c_4 SH_{\max}}{\Lambda\eta_{\min}^3(n-2K)^2},$$

$$\sqrt{\frac{2\beta}{T_{j,n}} \sum_x G_j(x)} \leq \sqrt{\frac{2\beta\Lambda}{n-2K}} + \frac{18\beta\sqrt{SH_{\max}}}{\eta_{\min}(n-2K)}.$$

Now, applying Lemma 13 yields

$$\frac{2\beta}{T_{k,n}} \sum_x G_k(x) \leq \frac{2\beta\Lambda}{n-2K} + \frac{37\beta\sqrt{SH_{\max}}}{n-2K} \sum_j \left( \sqrt{\frac{2\beta\Lambda}{n-2K}} + \frac{18\beta\sqrt{SH_{\max}}}{\eta_{\min}(n-2K)} \right)$$

$$+ \frac{312\beta SH_{\max}}{n-2K} \sum_j \left( \frac{\beta}{\eta_{\min}(n-2K)} + \frac{c_3}{\eta_{\min}^2(n-2K)^{3/2}} \sqrt{2SH_j/\Lambda} + \frac{4c_4 SH_j}{\Lambda\eta_{\min}^3(n-2K)^2} \right)$$

$$\leq \frac{2\beta\Lambda}{n-2K} + \frac{53\beta^{3/2}K\sqrt{SH_{\max}\Lambda}}{(n-2K)^{3/2}} + \frac{978KSH_{\max}\beta^2}{\eta_{\min}(n-2K)^2} + \widetilde{\mathcal{O}}((n-2K)^{-5/2}).$$

Finally, using the inequality $(n-2K)^{-1} \leq n^{-1} + 4Kn^{-2}$ and $n - 2K \geq n/2$ valid for all $n \geq 4K$, we get the desired result:

$$\frac{2\beta}{T_{k,n}} \sum_x G_k(x) \leq \frac{2\beta\Lambda}{n} + \frac{150\beta^{3/2}K\sqrt{S\Lambda H_{\max}}}{n^{3/2}} + \frac{3912KSH_{\max}\beta^2}{\eta_{\min}n^2} + \widetilde{\mathcal{O}}(n^{-5/2}).$$

$\square$

### D.3 Proof of Lemma 13

The proof borrows some ideas from the proof of Lemma 1 in [2]. Consider a chain $j$ that is sampled at least once after initialization, and let $t + 1 (> 2K)$ be the last time it was sampled. Hence,

$T_{j,t} = T_{j,n} - 1$. Using the same arguments as in the beginning of the proof of Lemma 12, we have on the events $C$ and $E$,

$$\frac{2\beta}{T_{k,n}} \sum_x G_k(x) \le \frac{1}{T_{j,n} - 1} \left( 2\beta \sum_x G_j(x) + c_3 \sqrt{\frac{8SH_j \sum_x G_j(x)}{T_{j,n}}} + \frac{8c_4 SH_j}{T_{j,n}} \right). \tag{10}$$

Note that (10) is valid for any $k$, and any $j$ that is sampled after initialization.

Now consider a chain $j$ such that $T_{j,n} - 2 \ge \eta_j (n - 2K)$. In other words, $j$ is over-sampled (w.r.t. budget $n - 2K$). In particular, $j$ is sampled at least once after initialization. Hence, using (10) and noting that $T_{j,n} \ge \eta_j(n - 2K) + 2$, we obtain

$$\frac{2\beta}{T_{k,n}} \sum_x G_k(x) \le \frac{1}{\eta_j(n - 2K)} \left( 2\beta \sum_x G_j(x) + c_3 \sqrt{\frac{8SH_j \sum_x G_j(x)}{\eta_j(n - 2K)}} + \frac{8c_4 SH_j}{\eta_j(n - 2K)} \right)$$

$$\overset{(a)}{\le} \frac{2\beta\Lambda}{n - 2K} + \frac{c_3\sqrt{8S\Lambda H_j}}{\eta_j(n - 2K)^{3/2}} + \frac{8c_4 SH_j}{\eta_j^2(n - 2K)^2}, \tag{11}$$

where (a) follows from the definition of $\eta_j$. Multiplying both sides on $\frac{\eta_k}{2\Lambda}$ gives:

$$\frac{\beta}{T_{k,n}} \le \frac{\beta}{\eta_k(n - 2K)} + \frac{c_3\sqrt{2S\Lambda H_j}}{\Lambda\eta_k\eta_j(n - 2K)^{3/2}} + \frac{4c_4 SH_j}{\Lambda\eta_k\eta_j^2(n - 2K)^2} \tag{12}$$

$$\le \frac{\beta}{\eta_{\min}(n - 2K)} + \frac{c_3}{\eta_{\min}^2(n - 2K)^{3/2}}\sqrt{2SH_j/\Lambda} + \frac{4c_4 SH_j}{\Lambda\eta_{\min}^3(n - 2K)^2},$$

thus verifying the first statement of the lemma. To derive the second statement, we take square-root from both sides of (11):

$$\sqrt{\frac{2\beta}{T_{k,n}} \sum_x G_k(x)} \le \sqrt{\frac{2\beta\Lambda}{n - 2K} + \frac{c_3\sqrt{8S\Lambda H_j}}{\eta_j(n - 2K)^{3/2}} + \frac{8c_4 SH_j}{\eta_j^2(n - 2K)^2}}$$

$$\le \sqrt{\frac{2\beta\Lambda}{n - 2K} + \frac{c_3\sqrt{8S\Lambda H_j}}{\eta_j(n - 2K)^{3/2}}} + \frac{\sqrt{8c_4 SH_j}}{\eta_j(n - 2K)}$$

$$\le \sqrt{\frac{2\beta\Lambda}{n - 2K}} + \frac{\sqrt{c_3/\beta}\sqrt{SH_{\max}}}{\eta_{\min}(n - 2K)} + \frac{\sqrt{8c_4 SH_{\max}}}{\eta_{\min}(n - 2K)},$$

where the second and third inequalities respectively follow from $\sqrt{a + b} \le \sqrt{a} + \sqrt{b}$ and $\sqrt{a + b} \le \sqrt{a} + \frac{b}{2\sqrt{a}}$ valid for all $a, b > 0$. Plugging $c_3 = 13\beta^{3/2}$ and $c_4 = 39\beta^2$ into the last inequality verifies the second statement and concludes the proof. $\square$

# E  Asymptotic Analyses — Proofs of Lemma 1 and Theorem 3

## E.1  Proof of Lemma 1

Consider a chain $k$, and let us denote by $\widetilde{P}_{k,n}$ the corresponding empirical estimator of $P_k$ (corresponding to $\alpha = 0$). That is, for all $(x, y) \in \mathcal{S}^2$, $\widetilde{P}_{k,n}(x, y) = \frac{1}{T_{k,x,n}} \sum_{t=2}^n \mathbb{I}\{X_{t-1} = x, X_t = y\}$. Further, let $\widetilde{L}_{k,n}$ denote the corresponding loss of $\widetilde{P}_{k,n}$ for chain $k$. To prove the lemma, we first show that $\lim_{T_{k,n} \to \infty} T_{k,n} L_{k,n} = \lim_{T_{k,n} \to \infty} T_{k,n} \widetilde{L}_{k,n}$.

To this end, we derive upper and lower bounds on $L_{k,n}$ in terms of $\widetilde{L}_{k,n}$. We have for all $(x, y) \in \mathcal{S}^2$:

$$|(\widehat{P}_{k,n} - \widetilde{P}_{k,n})(x, y)| = \left| \frac{\sum_{t=2}^n \mathbb{I}\{X_{t-1} = x, X_t = y\} + \alpha}{T_{k,x,n} + \alpha S} - \frac{\sum_{t=2}^n \mathbb{I}\{X_{t-1} = x, X_t = y\}}{T_{k,x,n}} \right|$$

$$= \frac{\alpha}{T_{k,x,n}(T_{k,x,n} + \alpha S)} \left| T_{k,x,n} - S \sum_{t=2}^n \mathbb{I}\{X_{t-1} = x, X_t = y\} \right|$$

$$\le \frac{\alpha S T_{k,x,n}}{T_{k,x,n}(T_{k,x,n} + \alpha S)} \le \frac{\alpha S}{T_{k,x,n}}.$$

We therefore get, on the one hand,

$$L_{k,n} \geq \sum_x \hat{\pi}_{k,n}(x) \|\widehat{P}_{k,n}(x,\cdot) - \widetilde{P}_{k,n}(x,\cdot)\|_2^2 + \widetilde{L}_{k,n} \geq \widetilde{L}_{k,n} \,,$$

and on the other hand,

$$L_{k,n} \leq \sum_x \hat{\pi}_{k,n}(x) \|\widehat{P}_{k,n}(x,\cdot) - \widetilde{P}_{k,n}(x,\cdot)\|_2^2 + \sum_x \hat{\pi}_{k,n}(x) \|P_k(x,\cdot) - \widetilde{P}_{k,n}(x,\cdot)\|_2^2$$

$$+ 2 \underbrace{\sum_x \hat{\pi}_{k,n}(x) \sum_y |\widehat{P}_{k,n}(x,y) - \widetilde{P}_{k,n}(x,y)| |\widetilde{P}_{k,n}(x,y) - P_k(x,y)|}_{A}$$

$$\leq \sum_x \hat{\pi}_{k,n}(x) \|\widehat{P}_{k,n}(x,\cdot) - \widetilde{P}_{k,n}(x,\cdot)\|_2^2 + \widetilde{L}_{k,n} + 2A$$

$$\leq \sum_x \frac{T_{k,x,n}}{T_{k,n}} \sum_y \left(\frac{\alpha S}{T_{k,x,n}}\right)^2 + \widetilde{L}_{k,n} + 2A = \frac{\alpha^2 S^3}{T_{k,n}} \sum_x \frac{1}{T_{k,x,n}} + \widetilde{L}_{k,n} + 2A \,.$$

Furthermore, $A$ is upper bounded as follows:

$$A \leq \sum_x \hat{\pi}_{k,n}(x) \frac{\alpha S}{T_{k,x,n}} \sum_y |\widetilde{P}_{k,n}(x,y) - P_k(x,y)|$$

$$\leq \sqrt{\sum_x \sum_y \hat{\pi}_{k,n}(x) \frac{\alpha^2 S^2}{T_{k,x,n}^2}} \sqrt{\sum_x \hat{\pi}_{k,n}(x) \sum_y (\widetilde{P}_{k,n}(x,y) - P_k(x,y))^2}$$

$$= \sqrt{\sum_x \frac{\alpha^2 S^3}{T_{k,n} T_{k,x,n}}} \sqrt{\widetilde{L}_{k,n}} \,,$$

where we have used Cauchy-Schwarz in the second line. In summary, we have shown that

$$\widetilde{L}_{k,n} \leq L_{k,n} \leq \widetilde{L}_{k,n} + \frac{\alpha^2 S^3}{T_{k,n}} \sum_x \frac{1}{T_{k,x,n}} + 2\sqrt{\sum_x \frac{\alpha^2 S^3}{T_{k,n} T_{k,x,n}}} \sqrt{\widetilde{L}_{k,n}} \,,$$

so that

$$T_{k,n} \widetilde{L}_{k,n} \leq T_{k,n} L_{k,n} \leq T_{k,n} \widetilde{L}_{k,n} + \alpha^2 S^3 \sum_x \frac{1}{T_{k,x,n}} + 2\sqrt{\sum_x \frac{\alpha^2 S^3}{T_{k,x,n}}} \sqrt{T_{k,n} \widetilde{L}_{k,n}} \,.$$

Taking the limit when $T_{k,n} \to \infty$ and noting the fact that when $T_{k,n} \to \infty$, by ergodicity, $T_{k,x,n} \to \infty$ for all $x \in \mathcal{S}$, we obtain $\lim_{T_{k,n} \to \infty} T_{k,n} L_{k,n} = \lim_{T_{k,n} \to \infty} T_{k,n} \widetilde{L}_{k,n}$.

It remains to compute $\lim_{T_{k,n} \to \infty} T_{k,n} \widetilde{L}_{k,n}$. We have

$$\widetilde{L}_{k,n} = \frac{1}{T_{k,n}} \sum_x T_{x,n} \sum_y (\widetilde{P}_{k,n}(x,y) - P_k(x,y))^2$$

$$= \frac{1}{T_{k,n}} \sum_x \sum_y \underbrace{[\sqrt{T_{k,x,n}}(\widetilde{P}_{k,n}(x,y) - P_k(x,y))]^2}_{Z(x,y)} \,.$$

When $T_{k,n} \to \infty$, by ergodicity, we have $T_{k,x,n} \to \infty$ for all $x \in \mathcal{S}$. Therefore, by the central limit theorem, $\sqrt{T_{k,x,n}}(\widetilde{P}_{k,n}(x,y) - P_k(x,y))$ converges (in distribution) to a Normal distribution with variance $[P_k(I - P_k)](x,y)$, and $Z(x,y)$ converges to a Gamma distribution with mean $[P_k(I - P_k)](x,y)$. Hence, the mean of $\widetilde{L}_{k,n}$ would be $\frac{1}{T_{k,n}} \sum_x \sum_y [P_k(I - P_k)](x,y) = \frac{1}{T_{k,n}} \sum_x G_k(x)$, thus completing the proof. $\qquad \square$

### E.2 Proof of Theorem 3

Let $n$ be a budget such that $\sqrt{n} \geq n_{\text{cutoff}}$, and let $n_0 := \sqrt{n}$. The proof proceeds in two steps. We first consider the case with budget $n_0$, and show that at the end of this sub-problem, the index of each chain is well estimated. Then, we consider allocation in the second sub-problem.

**Step 1: Bounds on the index at $t = n_0$.** Fix a chain $k$. To derive upper and lower bounds on the index $b_{k,t}$ at $t = n_0$, we derive a lower bound on $T_{k,n_0}$. Recall that by Lemma 9, we have with the choice $n = n_0$,

$$\frac{S\beta^2}{9T_{k,n_0}} \sum_{x:T_{k,x,n_0}>0} \frac{1}{T_{k,x,n_0} + \alpha S} \leq \frac{0.187 K S^2 \beta^2}{n_0 - K},$$

with probability at least $1 - \delta$. Using the trivial bound $T_{k,x,n_0} \leq T_{k,n_0}$, it follows that

$$\frac{S^2 \beta^2}{9T_{k,n_0}(T_{k,n_0} + \alpha S)} \leq \frac{0.175 K S^2 \beta^2}{n_0 - K},$$

so that

$$T_{k,n_0} \geq \sqrt{\frac{n_0 - K}{1.575 K}} - \frac{1}{3} \geq \frac{1}{4}\sqrt{\frac{n_0}{K}} \geq \frac{n^{1/4}}{4\sqrt{K}},$$

with probability greater than $1 - \delta$.

Noting that $n_0 \geq n_{\text{cutoff}}$, we apply Lemma 8 and 11 to obtain

$$\frac{2\beta}{T_{k,n_0}} \sum_x G_k(x) \leq b_{k,n_0+1} \leq \frac{2\beta}{T_{k,n_0}} \sum_x G_k(x) + \frac{19\beta^{3/2}\sqrt{S}}{T_{k,n_0}^{3/2}} \sum_x \sqrt{\frac{G(x)}{\pi_k(x)}} + \frac{90\beta^2 S}{T_{k,n_0}^2} \sum_x \frac{1}{\pi_k(x)},$$

with probability at least $1 - 2\delta$. Using Cauchy-Schwarz and recalling $H_k := \sum_k \pi_k(x)^{-1}$, we obtain give

$$\frac{2\beta}{T_{k,n_0}} \sum_x G_k(x) \leq b_{k,n_0+1} \leq \frac{2\beta}{T_{k,n_0}} \sum_x G_k(x) + \frac{19\beta^{3/2}}{T_{k,n_0}^{3/2}} \sqrt{SH_k \sum_x G_k(x)} + \frac{78\beta^2 SH_k}{T_{k,n_0}^2},$$

so that

$$\sum_x G_k(x) \leq \frac{T_{k,n_0}}{2\beta} b_{k,n_0+1} \leq \sum_x G_k(x) + \frac{10\beta^{1/2}}{T_{k,n_0}^{1/2}} \sqrt{SH_k \sum_x G_k(x)} + \frac{39\beta SH_k}{T_{k,n_0}},$$

with probability at least $1 - 2\delta$. Using the lower bound $T_{k,n_0} \geq \frac{n^{1/4}}{4\sqrt{K}}$ yields

$$\sum_x G_k(x) \leq \frac{T_{k,n_0}}{2\beta} b_{k,n_0+1} \leq \sum_x G_k(x) + \frac{20\beta^{1/2}}{n^{1/8}} \sqrt{SKH_k \sum_x G_k(x)} + \frac{156\beta SH_k \sqrt{K}}{n^{1/4}},$$

with probability at least $1 - 2\delta$. Let us write the last inequality as

$$\sum_x G_k(x) \leq \frac{T_{k,n_0}}{2\beta} b_{k,n_0+1} \leq \sum_x G_k(x) + \varepsilon_n,$$

where $\varepsilon_n = \widetilde{\mathcal{O}}(n^{-1/8})$, thus giving

$$\frac{2\beta}{T_{k,n_0}} \sum_x G_k(x) \leq b_{k,n_0+1} \leq \frac{2\beta}{T_{k,n_0}} \left( \sum_x G_k(x) + \varepsilon_n \right).$$

**Step 2: The second sub-problem.** Now let us consider $n_0 + 1 \leq t \leq n$. It follows that with probability $1 - 2\delta$, **BA-MC** allocates according to the following problem

$$\max_{\xi \in [0,\varepsilon_n]^K} \max_k \frac{2\beta}{x_k + T_{k,n_0}} \left( \sum_x G_k(x) + \xi_k \right) \quad \text{s.t.:} \quad \sum_k x_k = n - \sqrt{n},$$

whose optimal solution satisfies

$$\frac{(n - \sqrt{n}) \sum_x G_k(x)}{\Lambda + K\varepsilon_n} \leq x_k \leq \frac{(n - \sqrt{n}) \left( \sum_x G_k(x) + \varepsilon_n \right)}{\Lambda}.$$

Recalling $\varepsilon_n = \widetilde{\mathcal{O}}(n^{-1/8})$ and noting that $T_{k,n} \geq x_k$, we obtain $\frac{T_{k,n}}{n} \to_{n\to\infty} \frac{\sum_x G_k(x)}{\Lambda}$.

It remains to show that the loss $L_n(\textbf{BA-MC})$ approaches $\Lambda/n$ as $n$ tends to infinity. By Lemma 1, recall that $\lim_{T_{k,n}\to\infty} T_{k,n}L_{k,n} = \sum_x G_k(x)$. So, using $\frac{T_{k,n}}{n} \to_{n\to\infty} \frac{\sum_x G_k(x)}{\Lambda}$, we conclude $\lim_{n\to\infty} nL_{k,n} = \Lambda$, and the claim follows. $\qquad\square$