[Reviews · NeurIPS 2019]

Reviewer 1



The paper studies a problem of transition matrix estimation for Markov processes uniformly in a sequential resource allocation setting. The paper introduces relevant notions from the theory of Markov chains and the studied performance measure. The authors introduce the BA-MC algorithm based on the optimism in the face of the uncertainty principle. Then they prove bounds for asymptotic and non-asymptotic bounds with asymptotic performance achieving the optimal loss. Pros: * Detailed analysis for the static allocation scenario * Optimistic bound is based on a new concentration inequality for smoothed estimators * Results utilize both notions of Markov chain spectral gaps for reversible and non-reversible cases * BA-MC achieves the asymptotically optimal loss Cons: * It would be great to see any practical motivation for the setting, right now it feels like a purely theoretical construct * The performance criterion with true state distributions seems more relevant and less tailored for "convenience", although I agree that the results should carry over to this loss AFTER FEEDBACK: Thanks to the authors for the feedback. The paper was already in a good shape, but would benefit from including the mentioned points. I would still recommend acceptance.

Reviewer 2



This paper aims at learning a collection of transition matrices of ergodic Markov chains, where at each round the algorithm can select one of the chains and observe which state it fell in. The problem consists in designing a strategy such as the learning will occur uniformly over all chains at the best possible rate. The paper is of theoretical nature, the background on chains is properly introduced, the algorithm is clearly described and thoroughly analyzed. The paper in its current form is a stronger submission than its previous version. It is more focused, the assumptions are clearer, it is more detailed, and an overall better read. Bound on the loss is also provided in terms of simple quantities, namely the number of considered chains and number of states, which offers a priori guarantees under mild assumptions. The technical proofs in the appendix were not verified by the reviewer. === Some minor comments === line-30 indexing switches from n to m line-30 kernel is later defined to be row stochastic line-122 ’Moreover’ could imply that this is a byproduct of reversibility. However this is always true. line-140 summation should be over y line-143 At line 143 you make an assumption on the sum of the Gini indexes to be greater than 0. Interestingly, could you come up with an example of an ergodic chain for which this quantity would be null ? line-157 Indexes t’ and k of X should be switched for consistency. line-161,162 Notice that the estimator for the stationary distribution is not the stationary distribution of the estimator for the transition matrix. This could be recovered (if you somewhere in the proofs rely on that fact) by making the smoothing on your estimator of the stationary distribution coarser. line-279 Does it hold when both irreducibility and aperiodicity are dropped or do you have to keep one of these ? Remark 1 deserves to be made more precise. For example, are you still relying on other other facts, such as the positivity of the sum of Gini indexes ? ================ AFTER REBUTTAL ================== The reviewer thanks the authors for their detailed answers, believes that the authors will make the changes they promised and raises the score accordingly. About the reply at L.33-38 regarding the condition on the Gini indexes. The reviewer agrees that deterministic cycles would yield 0, but interestingly, isn't then your chain periodic hence not ergodic (under the definition that the transition matrix of an ergodic chain is primitive) ? But then, what sort of behavior would you observe if it's not a cycle but still has deterministic entries ?

Reviewer 3



The paper considers the novel problem of estimating the transition probabilities of K different Markov Chains uniformly well, given a bound on the total number of samples allowed to draw. This is a generalization of the well-studied problem where the goal is the same but with K real-valued distributions. The paper then provides an algorithm which is then analyzed in various regimes. This amalysis also shows that the algorithm is optimal in the asymptotic regime. The authors have done a really nice job in explaining the difficulties and the major steps both in the construction and in the analyis, making the paper an enjoyable read, and highlighting the main differences from the existing works also in this respect. The one thing that was missing for me is the motivation of this particular generalization of the classic problem on the real-valued distributions to the Markov chains. Nevertheless, the paper has some really nice ideas; in particular, I have found Lemma 1 relating the product of the losses and the sample sizes to the Gini index particularly interesting. (The second step in the proof summary could be a bit more detailed, however, to make it easier to understand why the result indeed makes sense.) In view of all this, I recommend accept. Additional remarks: lines 39 and 201: "rely on the Wald’s second identity" -> "rely on Wald’s second identity" line 140: presumably the sum is over y in S line 150: "according to P_{k_t}"? line 210: "denotes" -> "denote"

[Author Response · NeurIPS 2019]

We thank the reviewers for their constructive and positive comments. They will improve the quality of the paper.

**About motivation and potential practical applications (Reviewers 1 and 3).**   The study of the generalization of
this learning problem from real-valued distributions (i.i.d. setup) to Markov chains is interesting in itself from a
theoretical perspective: In contrast to our studied problem where various regimes appear as the budget varies, in the
i.i.d. case only a single regime exists. Markov chains have been successfully used for modeling a broad range of
practical problems, and their success makes "active learning in Markov chains" relevant in practice. Furthermore, there
are practical applications in reinforcement learning (RL) and in rested Markov bandits, for which our results could
prove beneficial. As an instance in RL, we mention the problem of "active exploration in MDPs" (see [28]), where the
task is to estimate the transition kernel of an unknown MDP uniformly well over state-action space, using a budget of $n$
samples. For the case of ergodic MDPs, each policy in the MDP defines an ergodic chain, and hence, the leaning task
becomes actively learning multiple Markov chains (we also note that compared to the setup in the present paper, active
learning in MDPs poses more challenges, as one has to consider a subset of all policies due to overlap among them.
However, we believe that our contribution could be beneficial for researchers in the RL community studying problems
related to active learning and exploration in MDPs). We may also refer to applications falling in the framework of
rested Markov bandits, for example channel allocation in wireless communication systems where a given channel's state
follows a Markov chain (e.g., Gilbert-Elliot channel model). Active learning in Markov chains is a relevant problem for
such applications, and we believe our contributions could serve as a technical tool for these applications. We agree to
strengthen the motivation of studying this problem and to widen the scope of the paper in view of this discussion.

**About the use of empirical stationary distribution in the loss function (Reviewer 1).**   The intention for using the
term "less meaningful" is partly illustrated in the paper (lines 207–218). We provide further detailed explanation
below, and agree to rewrite the corresponding part in Section 2.3, in view of the following discussion, so as to further
clarify the motivation of using $\hat{\pi}_{k,n}$. We aim to derive performance guarantees on the algorithm's loss that hold with
high probability (for $1 - \delta$ portions of the sample paths of the algorithm for a given $\delta$), as opposed to those holding
only in expectation. To this end, the loss $L_n$ (which uses $\hat{\pi}_{k,n}$) is more natural and meaningful than $L_n''$ (which uses
$\pi_k$; see line 189) as $L_n$ penalizes the algorithm's performance by the relative visit counts of various states in a given
sample path (through $\hat{\pi}_{k,n}$), and not by the expected value of these. This matters a lot in the small-budget regime
($n < n_{\text{cutoff}}$), where $\hat{\pi}_{k,n}$ could differ significantly from $\pi_k$ — Otherwise when $n \geq n_{\text{cutoff}}$, $\hat{\pi}_{k,n}$ is well concentrated
around $\pi_k$ with high probability. Reiterating the discussion in Section 2.3, let us consider the small-budget regime, and
some state $x$ where $\pi_k(x)$ is not small. In the case of $L_n$, using $\hat{\pi}_{k,n}$ we penalize the performance by the mismatch
between $\widehat{P}_{k,n}(x, \cdot)$ and $P_k(x, \cdot)$, weighted proportionally to the number of rounds the algorithm has actually visited $x$.
In contrast, in the case of $L_n''$, weighting the mismatch proportionally to $\pi_k(x)$ does not seem reasonable since in a
given sample path, the algorithm might not have visited $x$ enough even though $\pi_k(x)$ is not small.

**Minor comments.**   About chains with $\sum_x G_k(x) = 0$ (Reviewer 2): There exists ergodic chains with $\sum_x G_k(x) =$
$0$. The definition of the Gini index implies that such chains are necessarily deterministic (or degenerate), i.e. their
transition matrices belong to $\{0,1\}^{S \times S}$. One example is a deterministic cycle with $S$ nodes. So by assuming
$\sum_x G_k(x) > 0$, the analysis of Theorem 2 indeed excludes degenerate ergodic chains (satisfying $\sum_x G_k(x) = 0$). In
other words, the theorem is valid for *almost* all ergodic chains. We note however that the assertion of Theorem 1 still
holds even if $\sum_x G_k(x) = 0$. We will provide a footnote in page 7 to clarify this.

About estimator for empirical stationary distribution (Reviewer 2): This is indeed a nice remark. Our algorithm and
proofs do not rely on this fact, and we will include a remark on this in the paper. We also note that we use empirical
estimate $\hat{\pi}_{k,n}$ of $\pi_k$ in $L_n$ as it naturally corresponds to the occupancy of various states according to a given sample
path, and hence, its use can be intuitively justified.

About Remark 1 (Reviewer 2): The proof of Theorem 1 uses *entry-wise* concentration of $\widehat{P}_{k,n}$ around $P_k$, under the
event $C$ (which occurs with probability greater than $1 - \delta$); the proof does not rely on any *trajectory-wise* concentration.
As a result, the theorem is valid even if irreducibility and aperiodicity are dropped. Moreover, the proof does not use the
arguments in the proof of Theorem 2, which require $\sum_x G_k(x) > 0$. Hence, Theorem 1 is valid even for deterministic
ergodic chains for which $\sum_x G_k(x) = 0$. We agree to make Remark 1 more precise in view of this discussion.

About sketch proof of Lemma 1 (Reviewer 3): We explain the second step of the proof with more details.

Typos (all reviewers): We will fix typos. Thanks a lot for constructive comments!

[Meta-Review · NeurIPS 2019]

All reviewers felt that this is a well-executed paper with good writing and solid results, therefore clearly worthy of acceptance. The only general complaint was that the setting may have been somewhat poorly motivated, and the authors should consider providing an illustrative motivating example in the final version of the paper.